# Multi-Omics Profiling of Lipid Variation and Regulatory Mechanisms in Poultry Breast Muscles

**DOI:** 10.3390/ani15050694

**Published:** 2025-02-27

**Authors:** Hongyuan Zhang, Yaqi Dai, Jinxing Gu, Hongtai Li, Ran Wu, Jiyu Jia, Jingqi Shen, Wanli Li, Ruili Han, Guirong Sun, Wenting Li, Xiaojun Liu, Yinli Zhao, Guoxi Li

**Affiliations:** 1The Shennong Laboratory, Henan Agricultural University, Zhengzhou 450046, China; zhy2792710031@163.com (H.Z.); 17596905128@163.com (Y.D.); gujinxing028@163.com (J.G.); lihongtai1999@163.com (H.L.); wuranran23@126.com (R.W.); jiyujia2024@126.com (J.J.); hq767918@163.com (J.S.); rlhan@126.com (R.H.); grsun2000@126.com (G.S.); liwenting_5959@hotmail.com (W.L.); xjliu2008@hotmail.com (X.L.); 2Institute of Animal Science, Henan Academy of Agricultural Sciences, Zhengzhou 450002, China; liwanli605@163.com; 3College of Biological Engineering, Henan University of Technology, Zhengzhou 450001, China

**Keywords:** poultry, breast Muscles, lipid Variation, transcriptome, positively selected genes

## Abstract

In poultry production, muscle fat content plays a critical role in determining meat quality. Accurately characterizing the lipid molecular composition and profiles of breast muscles is essential for advancing both production practices and breeding strategies. However, existing lipid-related studies remain limited to a narrow range of poultry species, and the influence of artificial selection on lipid regulatory mechanisms remains poorly understood. In this study, we conducted comprehensive lipidomics, RNA-seq, and selective pressure analyses on the mature breast muscles of four representative poultry species: Arbor Acres (AA) broiler chickens, dwarf guifei chickens (AG), quails (AC), and pigeons (AD). The aim was to achieve precise molecular characterization and to elucidate the underlying regulatory mechanisms. Our findings revealed significant interspecies differences in lipid content, particularly in TG (triglycerides) and PC (phosphatidylcholines), with PC lipids emerging as key biomarkers for accurately distinguishing lipid profiles among the breast muscles of these species. Furthermore, genes involved in glycerolipid and glycerophospholipid metabolism, including PLA2G12A, LPCAT2, DGAT2, and PEMT, exhibited divergent expression patterns and experienced differential selective pressures across the four poultry species. These findings provide valuable experimental evidence to enhance our understanding of lipid molecular characterization in poultry breast muscles and offer insights into optimizing breeding strategies to improve meat quality traits.

## 1. Introduction

Poultry meat stands as a primary source of animal protein for dietary consumption [1]. China’s poultry meat production mainly includes the domestic chicken (*Gallus gallus*), quail (*Coturnix coturnix*) and pigeon (*Aplopelia Bonaparte*), among which the quail and pigeon were put into the category of traditional livestock and poultry in May 2020. Chicken meat, the second most significant meat product in China [2], boasted a national production of 1989.1 thousand tons in 2021. This production primarily stems from white-feathered broilers, constituting 45% of chicken meat production, and yellow-feathered broilers, making up 38% of chicken meat production. China’s meat pigeon industry commands over 80% of the global total, with a staggering 490 million commercial pigeons in 2020. Currently, the global quail industry collectively raises around 1 billion birds, while China’s annual meat quail output in 2020 was approximately 100 million birds. These poultry meats hold a significant market share in consumption, each distinct in their meat quality attributes, catering to diverse consumer demands. Given the consumer interest in meat quality, the pursuit of producing poultry meat of superior quality and flavor emerges as a new breeding focus. However, there remains a lack of detailed characterization and comparative studies on the meat quality attributes of these poultry. 

Muscle lipid traits, encompassing intramuscular fat (IMF) content and fatty acid composition [3], stand as pivotal factors influencing poultry meat quality [4,5]. Within skeletal muscle, lipids are stored as lipid droplets in the cytoplasm of myofibers and adipocytes [6]. IMF within muscle significantly impacts key properties of poultry meat such as tenderness, juiciness [5], flavor [5], and aroma [7]. Comparatively, quail [8] and pigeon [9] exhibit lower fat and cholesterol levels in terms of nutrient composition than chicken, yet the specific differential lipid molecules remain unknown. Presently, there is a wealth of studies focusing on phenotypic characterization of lipid traits in chicken meat. These studies have revealed variations in chicken IMF content across different breeds [10], within different muscle types of the same breed [11], and under the influence of various factors such as sex [12] and age [13]. Nevertheless, due to technical constraints, aspects such as flavor and aroma in chicken muscle cannot be precisely characterized [14], limiting the exploration of novel meat quality traits. In contrast, there exists a scarcity of phenotypic lipid trait characterization in quail and pigeon meat, particularly the detailed understanding of lipid traits and their distinctions from chicken meat in terms of lipid molecules. This gap hinders a comprehensive grasp of meat quality in these poultry and impedes the discovery of superior meat traits in poultry for genetic enhancement. Lipidomics, a high-throughput analytical technique rooted in liquid chromatography-mass spectrometry (LC-MS), proves instrumental in unraveling the lipid composition of organisms with precision, allowing for accurate quantification and identification of lipid molecules [15]. In recent years, this technique has been effectively employed in livestock and poultry studies to pinpoint key lipid factors influencing meat quality [16,17,18]. For instance, Mi et al. identified 47 potential lipid markers by scrutinizing the lipidomics of Taihe ostrich chickens [19]. Despite these advancements, the application of lipidomics in poultry remains relatively limited, leaving many poultry’ muscle lipid traits inaccurately characterized.

Muscle lipid traits represent quantitative traits that are under the joint regulation of microefficient polygenes and master genes [20]. Unveiling the genetic basis and molecular regulatory mechanisms behind avian muscle lipid-related traits stands as a focal point in genetic breeding research. Biologically, chickens and quails belong to the *Phasianidae* family, with chickens in the *Gallus* genus and quails in the *Coturnix* genus. Pigeons belong to the *Columbidae* family and are classified under the *Columba* genus. Biologically, domestic chickens and quails belong to the Ornithoptera Chickenaeidae family, with the former falling under the genus Procapra and the latter under the genus Quail. On the other hand, domestic pigeons are part of the Ornithoptera Doves family, specifically the genus Pigeon. These species have undergone extensive human domestication and artificial selection processes, evolving into poultry primarily for egg or meat production. Apart from the kinship distance between species, factors like domestication [21] and artificial selection [22,23] likely contribute to the variations in muscle lipid traits among these poultry, although the genetic underpinnings of these differences remain ambiguous.

The advent of modern histological and bioinformatics techniques has facilitated the identification of numerous genes linked to muscle lipid traits in domestic chickens. For instance, Ye [24] discovered 173 differentially expressed genes between two breeds through transcriptome profiling of skeletal muscle in sex-linked dwarf and normal breeds. Their findings unveiled that IMF deposition in sex-linked dwarf chickens is modulated by adipocytokines, insulin, and other downstream signaling pathways. Presently, an increasing number of studies are striving to pinpoint genes associated with differential lipids with more precision and elucidate the molecular mechanisms underlying lipid trait formation through integrated transcriptome and lipidome analyses [11,25,26]. However, these investigations predominantly focus on a few select domestic chicken breeds. In contrast to domestic chickens, there exists a paucity of research concerning muscle lipid-related histology in quail and pigeon. The absence of comparative transcriptome analyses and integrated transcriptome and lipidome studies further hampers efforts to delineate subtle differences in muscle lipid traits among diverse poultry and the potential regulatory mechanisms governing their development. The genetic basis of artificial selection influencing the formation of muscle lipid traits across avian breeds or species remains largely unexplored in existing studies.

The edible portions of poultry primarily consist of breast and leg muscles [27]. Among these, the poultry muscle holds significant importance in poultry meat production, contributing to approximately one-third of the total meat yield and serving as a key target tissue for meat quality assessment. However, there exists a gap in understanding the heterogeneity of lipid compositional traits within the breast muscle and the mechanisms governing its development in poultry species such as domestic chicken, quail, and pigeon. Studies have demonstrated that when comparing meat quality attributes among poultry with varying growth rates, especially those at different marketable ages, selecting the age of physiological maturity proves more effective than sampling based on chronological growth and development stages [28]. To shed light on the lipid composition characteristics of the breast muscle across different poultry and unveil the molecular mechanisms steering the formation of breast muscle lipid traits influenced by domestication or artificial selection, the current research conducted lipidomic and transcriptomic analyses on the breast muscles of four poultry, including broiler chicken, broiler quail, and broiler pigeon. Within the two domestic chicken breeds studied, the AA broiler breed stands out as a specialized white-feathered broiler variety subjected to intense artificial selection, characterized by rapid growth rates and efficient feed utilization [29]. In contrast, the dwarf guinea fowl represents a meat and egg variety with tender meat, featuring a lower degree of specialization selection compared to the AA broiler breed. Through this investigation, a precise assessment of the characteristics and diversity of breast muscle lipid traits among the four poultry was achieved. This study identified the genes and regulatory pathways associated with lipid composition variations in the breast muscles of four distinct poultry species. These findings not only address the existing differences in lipid profiles among common poultry species but also explore the potential impact of artificial selection pressure on meat quality. These discoveries offer hope for improving poultry meat traits through selective breeding.

## 2. Materials and Methods

### 2.1. Ethical Standards

Animal experimental procedures were conducted in accordance with the “Guidelines for the Care and Use of Laboratory Animals” (Ministry of Science and Technology of China, 2006). All experiments involved in this study were approved by the Animal Protection and Utilization Committee of Henan Agricultural University.

### 2.2. Test Animals and Sample Collection

To assess the meat quality characteristics of the breast muscles from four poultry species with varying growth rates and marketable ages, animals at the physiological maturity stage (marketable age) were selected for sampling. Specifically, 45-day-old AA broilers, 120-day-old dwarf guinea fowl, 70-day-old yellow quails, and 35-day-old white-feathered king pigeons were chosen for this study, with three healthy individuals selected for each species. The dwarf guinea fowl (referred to as AG) and AA broilers (referred to as AA) were obtained from the Yuanyang Poultry Germplasm Resource Farm at Henan Agricultural University. They were provided with ad libitum feed and water, following dietary guidelines outlined in the NRC 1994 [30] and the Chinese standard for chicken rearing (NY/T33-2004) [31]. The yellow quails (referred to as AC) and pigeons (referred to as AD) were procured from the market. Yellow quail (AC) and pigeons (AD) are produced from nearby farms, and the feeding standards meet the Chinese feeding standard for quail (DB34/T 1608-2021 [32]; Local Standards—National Standards Information Public Service Platform) and the Chinese feeding standard for pigeons (T/CAAA 038-2020 [33]; Technical Regulations for the Management of Meat Pigeon Breeding—National Digital Standards Pavilion), respectively. All animals designated for testing were slaughtered following a 12 h fasting period. The left breast muscle tissues were then harvested, swiftly frozen in liquid nitrogen, and stored at −80 °C for subsequent histological analyses.

### 2.3. Total RNA Extraction from Tissues

Referring to Dou et al.’s scheme [34], Trizol (Invitrogen, Carlsbad, CA, USA) reagent was used to extract total RNA from fresh breast muscle tissues of four kinds of poultry. A Nanodrop2000 (NanoDrop Products, Wilmington, NC, USA) was used to detect the concentration and purity of the RNA, agarose gel electrophoresis was used to detect the integrity of the RNA, and Agilent2100 (2100 Electrophoresis Bioanalyzer Instrument, Wilmington, NC, USA) was used to determine the RIN value to complete the construction of related libraries. The cdna library was quantified using TBS380 (Picogreen, Solarbio, Beijing, China) and mRNA fragmentation buffer was added. Fragments of fragmentation about 300 bp were separated by magnetic bead screening and sequenced using Illumina Novaseq 6000 platform (TaKaRa, Dalian, China). The sequencing strategy was PE 150, resulting in a short paired-end sequence reading of 151 bp.

### 2.4. Breast Muscles Tissue Lipidome Assay

#### 2.4.1. Lipid Extraction

First, place 50 mg of frozen sample into a 2 mL centrifuge tube, add 6 mm steel beads, and 680 µL of extraction solution (methanol:water = 2:5). Add 400 µL of MTBE to the tube and then grind the sample in a cryomill at −10 °C for 6 min at 50 Hz. Allow the sample to extract for 30 min at 5 °C and 40 KHz, followed by a 30 min rest period at −20 °C. Centrifuge the sample at 13,000× *g* at 4 °C for 15 min. Collect 350 µL of the supernatant in an EP tube and dry it under nitrogen. Add 100 µL of extraction solution (isopropanol:acetonitrile = 1:1), vortex for 30 s, and perform low-temperature ultrasonic extraction for 5 min at 5 °C and 40 KHz. Centrifuge for 10 min at 13,000× *g* and 4 °C. Extract 20 µL of the supernatant as the quality control (QC) sample. Transfer the remaining supernatant to injection vials with internal cannulae for analysis on the machine.

#### 2.4.2. Lipid Detection and Characterization

The Ultra High Performance Liquid Chromatography (UHPLC) system coupled with a Q Exactive HF-X mass spectrometer (Thermo Fisher, Waltham, MA, USA) was employed for lipid detection. A 2 μL injection volume and a column temperature of 40 °C were utilized in the analysis. Both positive and negative ion scanning modes were employed to capture mass spectral signals. The raw data obtained were processed and assessed using the lipidomics processing software Lipidsearch -v5.1 (Thermo Fisher, Waltham, MA, USA). Characteristic peaks were identified by querying the library. The mass spectral information obtained from both MS and MS/MS analyses was compared with a metabolic database to identify lipid metabolites based on secondary mass spectral matching scores.

#### 2.4.3. Molecular Characterization of Lipids

We used the LIPID MAPS database (https://lipidmaps.org/, accessed on 16 September 2024). According to the database used, lipid metabolites were classified into eight major classes of fatty acyls (FA), glycerol esters (GL), glycerophospholipids (GP), sphingolipids (SP), sterolipids (ST), isoprenoid lipids (PR), glycolipids (SL), and polyketides (PK), and 96 subclasses, and according to the degree of unsaturation of the carbon chains, lipids were classified into saturated fatty acyls (SFA), monounsaturated fatty acyls (MUFA), polyunsaturated fatty acyls (PUFA), and odd-numbered fatty acyls (OFA), unsaturated fatty acyls (PUFA) and odd fatty acyls (ODD).

#### 2.4.4. Differential Lipid Screening

Supervised orthogonal partial least squares discrimination analysis (OPLS-DA) was used to perform multivariate statistical analysis for each sample combination and calculate the variable important values in the projection (VIP); *t*-test was used for significant difference analysis. Significantly different lipids between groups were screened under the conditions of VIP > 1 and *p* < 0.05.

### 2.5. Breast Muscle Tissue Transcriptome Sequencing and Analysis

#### 2.5.1. Library Construction and Sequencing

Using the same breast muscle tissue samples as in the lipidomic assay, total RNA was extracted, and high-quality total RNA with a total amount of ≥1 µg, a concentration of ≥35 ng/μL, an OD260/280 ≥ 1.8, and an OD260/230 ≥ 1.0 was used for library construction. In this study, an Illumina TruSeqTM RNA Sample Prep Kit (Thermo Fisher, Waltham, MA, USA) was used for library construction according to the instructions. mRNA was enriched and fragmented by Oligo dT, and then cDNA was synthesized by reverse transcription. Then, the adaptor was ligated, and the target bands were recovered by 2.2% agarose gel after 15 cycles of PCR amplification; the library was amplified by TBS380 (Picogreen) quantification, amplified by bridge PCR on cBot to generate clusters, and sequenced using Illumina Novaseq 6000 platform. The sequencing strategy was PE 150 with a read length of 2 × 150 bp.

#### 2.5.2. Sequence Quality Control and Comparative Annotation

Sequencing raw data were processed using fastp -v0.23.4 (https://github.com/OpenGene/fastp, accessed on 9 August 2024) [35] software to eliminate sequences containing sequencing junction sequences, low-quality reads, sequences with a high N rate, and sequences that were too short in length, and to obtain clean reads, which were used for the subsequent bioinformatics analysis. Among them, AA broiler, dwarf guinea fowl (reference genome gallus.GRCg7b), and quail (reference genome GCA_016699485.1) were analyzed according to the program with the reference genome. The HISAT 2 (http://ccb.jhu.edu/software/hisat2/index.shtml, accessed on 19 August 2024) [36] software was used to compare the clean data (reads) after quality control with the reference genome to obtain the mapped data (reads); TopHat2 [37] software was used (https://ccb.jhu.edu/software/tophat/index.shtml, accessed on 25 August 2024) to compare and evaluate the results. The pigeon breast muscle transcriptome was analyzed using a reference-free genomic program. Using Trinity -2.15.1 [38] (https://github.com/trinityrnaseq/trinityrnaseq/wiki, accessed on 29 August 2024) software, clean reads were assembled from scratch to generate overlapping clusters (conting) and single sequences (singletion); subsequently, using the TransRate [39] (https://github.com/Blahah/transrate/releases, accessed on 5 September 2024) software, the assembled sequences were filtered and optimized and protein sequence redundancy in the sequences was removed using CD-HIT -v4.8.1 (https://github.com/weizhongli/cdhit/releases, accessed on 15 September 2024) [40] software. Finally, assembly evaluation was completed using BUSCO [41] (Benchmarking Universal Single-Copy Orthologs, http://busco.ezlab.org, accessed on 28 September 2024) software. All transcripts after assembly were compared with six databases, including NR, Swiss-Prot, Pfam, COG, GO, and KEGG, for transcriptome functional annotation.

#### 2.5.3. Identification of Homologous Genes

The protein sequences among the four poultry species were compared and clustered using OrthoFinder [42] (https://github.com/davidemms/OrthoFinde, accessed on 15 September 2024) software to obtain homologous genes among the four poultry species (inflation value = 2.0) and then the Markov clustering algorithm (MCL) (inflation value = 2.0) was used to remove collateral homologous genes and obtain single-copy direct homologous genes for subsequent analysis.

#### 2.5.4. Direct Homologous Gene Expression Analysis

RSEM [43] (http://deweylab.biostat.wisc.edu/rsem, accessed on 19 September 2024) software was used to count the number of reads of each single-copy immediate homologous gene. The TPM value of genes was calculated by million mapping fragments per kilobase (FPKM) ×100. The abundance of sequences in the CDS region of a gene can reflect the expression level of the homologous gene, and the higher the abundance, the higher the expression level of the gene. In this study, the CDS region sequences of each group of homologous genes were used as the reference sequences, and the TPM value of each gene’s CDS region sequence in each sample was counted as the expression level of that homologous gene [44]. Based on the expression matrix of the homologous genes, the correlation distance among the four poultry species was calculated, the divergence time of the species was calculated using PAML [45], and the gene expression evolutionary tree was constructed using the NJ method.

#### 2.5.5. Homologous Gene *ka/ks* Analysis

Based on Zhang’s description [46], *KaKs*_calculator was used to estimate the ratio (*Ka*/*Ks* value) of the nonsynonymous substitution number (*Ka*) of each nonsynonymous locus to the synonym substitution number (*Ks*) of each synonym locus based on the NG model, and selection analysis was performed for each lineal homologous gene [47]. Typically, genes with *Ka*/*Ks* > 1 are considered to be under positive selection pressure, and genes with *Ka*/*Ks* < 1 are considered to be under purifying selection [48,49]. GO and KEGG enrichment analysis of genes under selection was performed using David [50], and functional clusters were significantly enriched when *p* < 0.05.

### 2.6. Breast Muscle Transcriptome and Lipidome Correlation Analysis

#### 2.6.1. Intrinsic Correlation Analysis of Transcriptome and Lipidome

The two-way orthogonal partial least squares (O2PLS) method was used to evaluate the intrinsic relationship between the breast transcriptome and lipidome datasets by calculating the scores for each sample as well as the loading values for each gene and lipid using the OmicsPLS -v2.0 [51] statistical analysis software available in the R language package Correlation.

#### 2.6.2. Comparative Analysis of Differential Gene and Differential Lipid Enrichment Pathways

KEGG pathway annotation and enrichment analyses were performed on differential genes in the poultry muscle transcriptome and differential lipids in the lipidome, respectively, and the VennDiagram package in R was used to compare the enriched pathways and identify the pathways in which differential genes and differential lipids were jointly involved [52]. In addition, differential genes and differential lipids were simultaneously mapped to pathways that were jointly involved, and the pathway data were integrated to visualize the KEGG pathway [25].

## 3. Results

### 3.1. Overview of Breast Muscle Lipid Composition in Four Poultry

A total of 1542 lipid molecules were identified in the breast muscles of the four poultry, detected in positive and negative ion modes (Appendix A). These lipids were classified into 50 subclasses under five major classes: fatty acyls (FAs), glycerolipids (GLs), glycerophospholipids (GPs), sphingolipids (SPs), and steroids (STs). Notably, the detected lipids were notably absent of isoprenoid lipids, glycolipids, and polyketides. Among these lipid classes, glycerophospholipids (GPs) accounted for 50.65% and glycerolipids (GLs) for 36.12% of the total lipids. The dominant subclasses within these major classes were phosphatidylcholine PC (13.22%), phosphatidylethanolamine PE (10.05%), and triglyceride TG (28.17%). This distribution is illustrated in Figure 1A.

In the analysis of chain length distribution in the breast muscles of the four poultry, a total of 67 lipid molecules were detected, showcasing a wide range of chain lengths from 8 to 86 carbon atoms. The prevalent presence of long-chain and ultra-long-chain fatty acids was observed, with the most abundant lipid molecules possessing chain lengths of 18, 16, 36, 40, 38, 20, 34, 42, 52, and 44, collectively representing 58.85% of the total molecules (as depicted in Figure 1B). Regarding the unsaturation of the identified lipid molecules, it was noted that 84.11% of the lipid molecules contained double bonds. The majority of these double-bonded lipids had 1 to 4 double bonds, accounting for 70.51% of all double-bonded lipid molecules. Lipid molecules with higher numbers of double bonds, such as 9, 11, 12, and 14, were less prevalent. In terms of lipid chain unsaturation, the analysis revealed a high level of lipid unsaturation in the avian poultry muscles, with polyunsaturated fatty acids (PUFA) constituting the highest proportion at 40.47%, followed by saturated fatty acids (SFA) at 23.28%, monounsaturated fatty acids (MUFA) at 18.75%, and other double-bonded lipids (ODD) at 17.5% (as shown in Figure 1D). The distribution of specific fatty acid types within these categories highlighted that MUFA were predominantly composed of 16:1, 18:1, 20:1, 14:1, and 30:1; PUFA were mainly represented by 18:2, 20:4, 14:2, 20:3, and 22:4; and SFA were primarily composed of 18:0, 22:0, 14:0, 16:0, and 20:0.

### 3.2. Comparison of Differential Lipid Profiles of Breast Muscles of Four Poultry

The results from correlation and PCA analyses of each sample indicate distinctive lipid profiles within the breast muscles of the four poultry, highlighting significant differences in lipid composition (refer to Figure 2A,B). Expression analysis revealed 711 differing lipid molecules in the breast muscles across the four poultry (*p* < 0.05 and VIP > 1). Among them, the least variation was observed between dwarf guinea fowl and quail, while pigeons exhibited a higher number of differential lipids compared to the other species (Figure 2C). These distinct lipid molecules were categorized into major classes such as fatty acids (FA), glycerolipids (GL), glycerophospholipids (GP), sphingolipids (SP), and sterol lipids (ST). Notably, GP and GL accounted for the highest numbers, with 412 and 195 molecules, respectively. Further subcategorization into 44 subclasses including DG, Lyso-phosphatidylglycerol (LPG), Phosphatidic acid (PA), and PC revealed TG, PC, PE, Cer, and DG as the most prominent subclasses (Figure 2D). The PCA emphasized a significant separation within the TG and PC subclasses, indicating their substantial contribution to the interspecies differences in poultry muscle lipid compositions among the poultry. Notably, the principal component analysis of the top 20 differential lipid molecules showcased PC as the subclass with the highest number of distinctive lipids, exhibiting a clear separation pattern. The top 10 lipids with notable separation included PC (22:1/14:1), PC (16:0e/18:2), PC (14:0e/22:5), PC (18:1/18:2), PE (18:0/20:4), PC (18:1/20:4), PC (16:0e/20:4), LPC (18:0), and CL (24:0/18:2/18:2/18:2) (Figure 2E). These findings underscore the significant interspecific differences in the differential lipid molecules present in the breast muscles of the four poultry.

Based on the analysis of differentiated lipid molecules in terms of carbon chain length and unsaturation characteristics, the predominant carbon chain lengths of differentiated lipids in AA broilers, dwarf guinea fowls, quails, and pigeons are mainly concentrated around 12, 14, 16, 18, 20, and 22 carbon atoms, showing no significant differences (Figure 2G). The highly expressed lipids in each poultry species were designated as dominant differentiated lipids. Analysis of unsaturation characteristics revealed that the ratio of polyunsaturated fatty acids (PUFA) to saturated fatty acids (SFA) in the dominant differentiated lipids was 1:1 in AA broilers (Figure 2H), while in AC and AG, this ratio was closer, at 1:0.82 and 1:0.87. In contrast, pigeons showed a PUFA:SFA ratio of 1:2, significantly differing from the composition ratios of the other three poultry species (refer to Figure 2F). In terms of expression levels and distribution, the dominant differentiated lipids in dwarf guinea fowls and quails were primarily from the triglyceride (TG) subclass, while in AA broilers, they were predominantly from the phosphatidylcholine (PC) subclass, and in pigeons, they were mainly from the lysophosphatidylcholine (LPC) subclass.

### 3.3. Comparative Analysis of Poultry Muscle Transcriptome

Using AA broiler, dwarf guinea fowl, quail, and pigeon poultry muscle tissues, a total of 12 cDNA libraries were constructed, and Raw reads of 162,237,762, 151,144,252, 142.830,722, and 156,656,292 were obtained by RNA-seq, and the sequences were quality assessed and processed to obtain 155,345, 574, 145,135,312, 136,520,842, and 151,681,272 clean reads, respectively. PCA showed that the poultry muscle samples of each poultry were clustered separately, with a good state of segregation between species (Figure 3A). A total of 5670 orthologous genes were identified from among the four poultry by comparison search (Appendix A). According to the evolutionary tree analysis, the closest evolutionary distance was between AA broiler and dwarf guinea fowl, followed by dwarf guinea fowl and quail, and pigeons maintained a long evolutionary distance from all other three poultry species (Figure 3B).

The expression analysis revealed 1012 differentially expressed orthologous genes between AA broilers and dwarf guinea fowl, 1586 between the two chicken breeds and quail, and 1336 between pigeons and chickens and quail. Notably, 66 orthologous genes were differentially expressed among all four poultry (Figure 3C, Appendix A). The four avian-specific dominant homologous genes were categorized based on their expression in the four poultry. Enrichment analysis results indicated that the functions of these dominant genes varied among the four poultry (Figure 3D). Specifically, the dominant homologous genes in AA broilers and pigeons were primarily associated with macromolecule catabolic processes, protein catabolic processes, and protein hydrolysis involved in protein catabolism (proteinolysis). Dominant homologous genes in dwarf guinea fowl were mainly linked to lipid-β oxidation, fatty acid β oxidation, lipid oxidation, and other lipid metabolic processes. In quail, the dominant homologous genes were predominantly related to lipid-β oxidation and other lipid metabolic processes, as well as fatty acid β oxidation and lipid metabolism. Moreover, dominant homologous genes in quail were primarily associated with RNA metabolism, including processes such as ncRNA metabolism, ribosome biogenesis, and rRNA metabolism. In addition, the dominant genes in all four poultry of quail were mainly involved in fatty acid β oxidation, lipid oxidation, lipid metabolism, and related pathways. Additionally, 66 commonly differentially expressed orthologous genes in the breast muscles of the four poultry were enriched. These genes were mainly implicated in threonine metabolic processes, aspartate family amino acid catabolic processes, organic acid metabolic processes, carboxylic acid catabolic processes, amino acid metabolism, carbohydrate metabolism, lipid metabolism, and other metabolism-related pathways (Figure 3E).

### 3.4. Adaptive Evolutionary Analysis of Homologous Genes

The adaptive evolutionary analysis of the orthologous homologous genes among the four poultry has been completed. The values of *ka*, *ks*, and *ka/ks* were calculated for each homologous gene, revealing an average of 7.6 synonymous substitutions and 6.4 nonsynonymous substitutions in the homologous genes. To ensure the accuracy of subsequent analyses, orthologous genes with *Ka/Ks* = 999 or 0.0001 in any one species were excluded. A total of 967 orthologous genes subject to positive selection (*Ka/Ks* > 1) were ultimately identified in the four poultry. Among these, 763 were in AA broilers, 767 in dwarf guinea fowl, 24 in quail, and 8 in pigeons (Appendix A). The number of genes under positive selection was significantly lower in pigeons and quail compared to domestic chicken breeds. The distribution of *Ka/Ks* values revealed that in pigeons, the range was 1.04151 to 2.31199. In quails, aside from ZNF746 (*Ka/Ks* = 421.342), MXRA7 (*Ka/Ks* = 356.432), CEP192 (*Ka/Ks* = 300.167), and HAUS1 (*Ka/Ks* = 19.2921), genes were subject to strong positive selection, with 83% of genes having *Ka/Ks* values ranging from 1 to 5. In AA broilers and dwarf guinea fowl, 14.8% and 16.6% of the positively selected genes had *Ka/Ks* ≥ 100, while 48.5% and 52.7% had 10 ≤ *Ka/Ks* < 100, respectively. This indicates that a majority of genes in the domestic chicken breeds, AA broilers, and dwarf guinea fowl underwent strong positive selection, contrasting with quail and pigeon genes, which experienced lower degrees of positive selection, aligning with their domestication history and levels of artificial selection they encountered. Figure 4C shows the enrichment of these genes.

The distribution of the 967 genes subjected to positive selection in the four poultry was further comparatively analyzed (Figure 4D). It was observed that only the ZNF746 (zinc finger protein 746, transcript variant X1) gene was positively selected in AA broilers (*Ka/Ks* = 34.9513), dwarf guinea fowl (*Ka/Ks* = 104.87), quail (*Ka/Ks* = 1.04851), and pigeons (*Ka/Ks* = 1.9456). This gene was universally subjected to positive selection across all four species. In pigeons, there were no genes specifically under positive selection compared to domestic chicken breeds, except for the ERMARD (ER membrane-associated RNA degradation, transcript variant X2) gene, which was also under positive selection in AA broilers or dwarf guinea fowl. However, in the comparison between pigeons and quail, there were minimal genes co-opted under positive selection between the two species, except for ASB10 (ankyrin repeat and SOCS box containing 10, transcript variant X1) and ERMARD. The shared gene, ERMARD, was also under positive selection in quail (*Ka/Ks* = 421.342) and pigeons (*Ka/Ks* = 1.41589), exhibiting significant differences in the degree of positive selection.

Furthermore, in quail, apart from TAF1B (TATA-box binding protein-associated factor, RNA polymerase I subunit B), CCDC25 (coiled-coil domain containing 25, transcript variant X2), and OAZ1 (ornithine decarboxylase antizyme 1) as specifically selected genes, the remaining selected genes were also positively selected in AA broilers or dwarf guinea fowl. These findings suggest potential greater divergence in the selection of muscle-related traits between pigeons and quail during domestication and artificial selection. In contrast, there appears to be more convergence in the selection of certain muscle-related traits between pigeons or quail and domestic chicken breeds.

The genes under positive selection and their functional enrichment were comparatively analyzed in two domestic chicken breeds. It was discovered that a total of 25 genes were positively selected between domestic chickens and either quail or pigeons, with close associations to body immunity or resistance (Figure 4E). Between AA broilers and dwarf guinea fowl breeds, 551 genes were found to be positively selected, primarily linked to the metabolism of various substances. Additionally, these genes exhibited more enriched pathways related to immunity and lipid metabolism (Figure 4F). Furthermore, 192 genes were exclusively subjected to positive selection in AA broilers, predominantly associated with DNA replication and repair, lipid metabolism, and cell cycle processes, among others. These genes showed more enrichment in pathways related to immunity and resistance (Figure 4G). Similarly, 195 genes were specifically under positive selection in dwarf guinea fowl, primarily involved in metabolism and immunity, with a higher enrichment of pathways related to lipid metabolism (Figure 4H).

These findings indicate that the artificial selection processes in domestic chickens, encompassing fast-growing meat breeds and parthenogenetic local breeds, have undergone considerable differential selection concerning basal metabolism, lipid metabolism, and immune resistance.

### 3.5. Lipid Metabolism-Related Genes Subject to Positive Selection

Based on the results of functional enrichment analysis, 237 selected genes related to lipid metabolism were identified from a pool of 967 immediate homologs. Among these, 114 genes were under positive selection (Appendix A), while 123 were under purifying selection. These positively selected genes involved in lipid metabolism participate in various biological processes and pathways related to lipid metabolism (Figure 5A). Notably, well-known lipid metabolism-related genes like DGAT2, DGKE, and LPL were also detected, indicating varying levels of selection pressure on these genes across the four poultry. Moreover, combining the results of differential expression analysis of immediate homologous genes, it was observed that 82 of the lipid metabolism-related genes under positive selection exhibited significant differential expression in the breast muscles of AA broilers and dwarf guinea fowl. In contrast, there were no lipid metabolism-related differentially expressed immediate homologous genes under positive selective pressures between quail and pigeons. Functional enrichment analysis revealed that these differentially expressed lipid metabolism-related genes under positive selection were predominantly enriched in biological processes such as glycerolipid and glycerophospholipid metabolism. Additionally, pathways such as glycerophospholipid metabolism and glycerolipid metabolism were notably enriched.

### 3.6. Poultry Muscle Lipidome and Transcriptome Association Analysis

To elucidate the molecular mechanisms underlying differences in lipid deposition in the muscles of four poultry, lipidomic and transcriptomic association analyses were conducted. The results demonstrated that the differential genes and lipids among the four poultry were significantly enriched in pathways related to lipid metabolism, including glycerophospholipid metabolism, sphingolipid degradation, and linoleic acid metabolism (Appendix A). This aligns with previous lipidomic findings that TG and PC-like differential lipids serve as primary markers for distinguishing lipid composition differences in the breast muscles of these poultry. However, the genes involved in PC, LPC, and TG-like lipid metabolism differed significantly across the four poultry (Appendix A).

In the generation of PC and LPC, genes such as LCAT, PLA2G6, PLA2G12A, and LPCAT2 play crucial roles. Their expression patterns and the selective pressures they undergo varied significantly among the poultry (Figure 6A). Specifically, LCAT and LPCAT2 were pivotal in influencing PC and LPC-like lipogenesis in domestic chicken AA broilers and dwarf guinea fowl, with LPCAT2 experiencing strong positive selection exclusively in AA broilers (*ka/ks* = 6.4125). This indicates a key role for LPCAT2 in the conversion of LPC-like lipids to PC-like lipids in the breast muscles of fast-growing broilers. While only PLA2G12A was differentially expressed in broilers and quail, the presence of LCAT, PLA2G6, and PLA2G12A between dwarf guinea fowl and quail impacted PC- and LPC-like lipogenesis across species. PLA2G12A exhibited varying levels of selection pressure among AA broilers (*ka/ks* = 0.31531), dwarf guinea fowl (*ka/ks* = 7.45521), and quail (*ka/ks* = 0.07958), with the highest expression in quail breast muscles compared to domestic chickens. Correspondingly, the differential lipid PC (15:0/18:2) content mirrored the expression levels of PLA2G12A among the three poultry. Moreover, species-specific differential lipid molecules, like LPC(16:0) exclusively in the AA broilers and quail comparison group, and LPC(20:5) solely in the dwarf guinea fowl and quail comparison group, were identified. These findings are likely linked to the expression patterns and selection pressures of the aforementioned genes among the species.

In the regulation of triglyceride (TG) generation, we observed significant differences in the expression pattern and selection pressure of the DGAT2 gene across the four poultry. Specifically, DGAT2 underwent positive selection in AA broilers (*ka/ks* = 1.00395) and neutral or purified selection in dwarf guinea fowl (*ka/ks* = 0.0001) and quail (*ka/ks* = 0.0620515). The expression of the DGAT2 gene in the breast muscle was highest in quail and dwarf guinea fowl, with no significant difference between these two species, while AA broilers exhibited the lowest expression levels. Correspondingly, the contents of the differential lipid molecules TG (16:1/18:2/20:4) and TG (18:2/20:4/20:4) displayed a similar expression pattern among the species (Figure 6C).

In contrast to the other three poultry, the metabolic pathways of phosphatidylcholine (PC), lysophosphatidylcholine (LPC), and triglycerides (TG) in pigeon breast muscle showed significant differences, with the related genes not following the aforementioned pattern of change or positive selection pressure. In the glycerophospholipid metabolism pathway, we observed that the PTDSS1 gene lost its involvement in PC catabolism in pigeons. However, other PC-generating pathways, notably involving the PEMT gene, played a crucial role (Figure 6A).

Furthermore, a network diagram illustrating the correlations between lipid metabolism-related differential genes subject to positive selection and breast muscle differential lipids in the four poultry was constructed (Figure 6B). The analysis revealed that genes such as MRPL32, PNPLA2, and MICALL1, which underwent positive selection exclusively in AA broilers, along with genes like CUL2, RAD51D, and ALDH7A1 subject to positive selection in Dwarf Guineas, and genes including RCAN1, SPTDSS2, and SSR2 under positive selection in both AA broilers and Dwarf Guineas, were significantly correlated with various differential lipids. Additionally, differential lipid molecules associated with key genes such as LCAT, PLA2G6, PLA2G12A, LPCAT2, PEMT, and DGAT2, which impact the metabolism of phosphatidylcholine (PC), lysophosphatidylcholine (LPC), and triglycerides (TG) in the breast muscles of the four poultry and exhibit distinct expression patterns, were identified (Appendix A). The analysis highlighted a strong negative correlation between PLA2G12A and the differential lipid PC (15:0/18:2) (Pearson = −0.99968) and a strong positive correlation between DGAT2 and the differential lipids TG (16:1/18:2/20:4) and TG (18:2/20:4/20:4) (Pearson = 0.98682, Pearson = 0.95868) (Figure 6C). This implies that these differential gene–lipid pairs could play a crucial role in influencing the lipid traits of the breast muscles in the four avian species.

## 4. Discussion

Muscle lipid traits are intricately linked to poultry meat quality, yet most poultry species lack precise characterization of these traits. In this study, ultra-high performance liquid chromatography (UHPLC) coupled with mass spectrometry was employed to conduct a lipidome analysis of the breast muscle tissues of four bird species: AA broiler, dwarf guinea fowl, quail, and pigeon. A total of 1542 lipid molecules across 50 subclasses were identified. Previous studies on the lipid composition of common poultry breast muscle mainly focused on the comparison of breast muscle of a single breed of poultry or the expression of related genes, but there was still a lack of comparative analysis between different birds and differences in regulatory mechanisms. The analysis revealed that the breast muscle lipids of these birds primarily comprised two major classes: glycerophospholipids (GP, 50.65%) and glycerol esters (GL, 36.12%), with subclasses such as triglycerides (TG), phosphatidylcholine (PC), phosphatidylethanolamine (PE), and other subclasses being predominant. Previous studies on the lipid composition of chicken [53] pigeon [8], and quail [9] breast muscles have also highlighted this characteristic. Glycerophospholipids are recognized as the major lipid component of biological membranes, crucial for the structure and function of all animal cells, membrane formation, and cell signaling [54]. PC is the main source of second messenger DAG, phosphatidic acid, lysophosphatidic acid, and arachidonic acid, and can be further metabolized into other signaling molecules, and is the main lipid molecule in meat that affects meat quality. [55] Current research shows that PE lipids can improve the elastic shrinkage rate of meat, while increasing the absorption capacity of water and nutrients, thereby improving the taste and flavor of meat [56]. As a key component representing intramuscular fat content, TG content changes have an important impact on meat quality [57]. The change in the content of these lipid molecules has a great influence on the meat quality. The dominant lipid compositions in avian breast muscles align with the physiological functions of these lipids. Moreover, this study delved into the transcriptomic profiles of the breast muscles of the four avian species, identifying 5790 direct homologous genes. This elucidation of the genetic foundation sheds light on the impact of domestication and artificial selection on lipid traits in the breast muscles of these avian species. This groundbreaking study represents the first cross-species comparative analysis of the lipidome and transcriptome of the breast muscles of these common avian species. The findings serve as valuable resources for further investigations into the regulatory mechanisms governing the formation of lipid-related quality traits in avian breast muscles.

In our comparative analysis of the lipid composition in the breast muscles of four avian species, we observed relatively minor differences in the lipid profiles of AA broilers, dwarf guinea fowls, and quails, with the polyunsaturated fatty acids (PUFA) to saturated fatty acids (SFA) ratio in their breast muscles being close to 1:1. In contrast, broiler pigeons exhibited distinct lipid composition characteristics, with higher levels of SFA compared to the other three species (Figure 2H). From a biological standpoint, domestic chickens and quails belong to two genera within the chicken family of the Chickeniformes order and are closely related. Domestic pigeons, on the other hand, belong to the pigeon genus in the Pigeonidae family and are more distantly related to domestic chickens and quails [58]. At the molecular level, studies by Zhou Xun et al. have shown that chickens and pigeons belong to different evolutionary clades based on whole-genome nucleotide sequence data from 47 bird species [58]. Wu et al. also demonstrated that quails share a closer evolutionary relationship with chickens, diverging approximately 22.2 million years ago [59]. Therefore, the observed differences in the lipid composition of the breast muscles of the four avian species in our study align with their taxonomic speciation, reflecting the proximity of their relationships. Interestingly, we also noted that quails and dwarf guinea fowls exhibited more similar breast muscle lipid compositions compared to AA broilers. Likewise, at the molecular level, gene expression patterns in the breast muscles of dwarf guinea fowls and quails were remarkably alike, with the evolutionary tree showing these two species as the most closely related among the four avian species (Figure 3B). We hypothesize that this similarity may be attributed to artificial directional selection during breed genetic improvement. AA broilers, a prevalent fast-growing commercial white-feathered broiler breed, have undergone intensive selection for growth traits. Studies by MJ et al. have demonstrated that commercial broiler chickens increased their growth rate by over 400% between 1957 and 2005 through specialized directional selection, enabling current fast-growing broilers to reach finishing weight in 40–60 days [60]. In contrast, dwarf guinea fowls, an egg–meat breed, have experienced less intense directional specialized selection, while meat quails have undergone selection comparable to dwarf guinea fowls. Our analysis of selection pressure on directly homologous genes revealed numerous genes specifically under positive selection in AA broilers and dwarf guinea fowls, respectively, with a significantly higher number of genes under positive selection in the breast muscles of dwarf guinea fowls and quails compared to AA broilers. Functional enrichment analysis indicated that genes under positive selection in the two domestic chicken breeds were primarily associated with various substance metabolisms, whereas genes specifically under positive selection in each breed were more linked to lipid metabolism. These findings suggest that, apart from breed factors, artificial directional selection likely plays a key role in shaping differences in breast muscle lipid traits among the four avian species. Glycerophospholipids are important components of the cell membrane and are also the intermediate molecules of intracellular signal transmission. In terms of lipid transport [61], glycerophospholipids can be used as lipid transport carriers and participate in the synthesis, processing, and transportation of lipids. As an important part of adipose tissue, glycerin is involved in energy metabolism and fat storage. In terms of lipid transport, glycerides play a key role in the process of fat deposition and decomposition [62]. Metabolism simultaneously affects lipid transport and oxidation. The metabolism and regulation of glycerol and phospholipid affect the stability of fat content in poultry meat.

Our focus was on analyzing the distinct lipid molecules present in the breast muscles of four avian species to uncover the reasons behind variations in their meat properties. Clustering the 711 differentiated lipid molecules (*p* < 0.05 and VIP > 1) revealed five major groups: fatty acids (FA), glycerolipids (GL), glycerophospholipids (GP), sphingolipids (SP), and sterols (ST). Notably, the GP group contained the highest number of differentiated lipid molecules, a trend observed in pigs [63], cattle [64], sheep [18], geese [65], and other domestic chicken breeds [53], emphasizing the critical role of GP in livestock and poultry meat quality. Principal Component Analysis (PCA) further highlighted that triglycerides (TG) and phosphatidylcholine (PC) subclasses exhibited significant segregation, contributing significantly to the differences in lipid composition among the breast muscles of the avian species. Particularly, the PC subclass displayed the highest number of differential lipid molecules and most pronounced segregation, with carbon chains primarily concentrated at 16 and 18 carbon atoms—lengths that align with optimal fatty acid chain lengths for birds [66], including saturated and unsaturated fatty acids like 16:0, 18:0, 16:1, 18:1, 16:2, 18:2, and 18:3. Regarding content, TG levels in AA broiler breast muscles were notably lower than in the other species, while PC was down-regulated in dwarf guinea fowl compared to AA broilers and in quails relative to domestic chickens. This indicates that TG and PC likely play crucial roles in determining meat quality in the breast muscles of these avian species, serving as key lipid molecules for assessing their lipid composition characteristics [67] (Appendix A). To delve deeper into the mechanisms driving differences in lipid composition, we conducted a combined lipidomic and transcriptomic analysis. We discovered significant enrichment of differential lipids and genes in the glycerophospholipid metabolism and glycerolipid metabolism pathways, particularly involving PC, LPC, and LPC, and PC, LPC, and LPC. These pathways were closely linked to the molecular metabolic processes of PC, LPC, and TG. Furthermore, our functional enrichment analysis of differentially expressed genes associated with lipid metabolism under positive selection highlighted their significant enrichment in pathways related to glycerophospholipid metabolism and glycerolipid metabolism. These findings align with the observed patterns of differential lipid distribution in the breast muscles of the avian species, indicating that glycerol lipid metabolism and glycerophospholipid metabolism pathways play pivotal roles in regulating the production of PC and TG—key differential lipids influencing lipid composition variations in the breast muscles of these avian species.

To unveil the potential molecular mechanisms underlying the impact of domestication or artificial selection on the lipid composition of the breast muscle in four poultry species, our focus was on pinpointing genes linked to the production of phosphatidylcholine (PC) and triglyceride (TG)-like lipid molecules within the glycerolipid metabolism and glycerophospholipid metabolism pathways. Studies with PC, which serves as a reservoir of polyunsaturated fatty acids (PUFAs), have shown that during storage, the content of PC is related to the oxidation rate of polyunsaturated fatty acids in meat, which will significantly affect the preservation ability of meat, and has been demonstrated to hold greater significance in elucidating variations in the proportion of fatty acid composition in the breast muscle across these avian species [68]. Our investigation identified LCAT, PLA2G6, PLA2G12A, LPCAT2, and PTDSS1 genes within the glycerophospholipid metabolism pathway that are closely associated with PC metabolism in the breast muscles of AA broilers, dwarf guinea fowl, and quail, showcasing evident interspecies differences in their expression patterns and levels of selection pressure. Among these, PLA2G12A, an enzyme belonging to the secreted phospholipase A2 (sPLA2) family, is capable of generating various lipid mediators that play pivotal roles in lipolytic and phospholipid catabolic processes, thereby enhancing the clearance of circulating triglycerides and the uptake of fatty acids by the liver [69]. In agricultural animals, this gene has been linked to intramuscular fat deposition exclusively in pigs, though no associated lipid molecules have been identified [70]. In our current research, we have identified intriguing insights regarding the genetic regulation of lipid composition in the breast muscles of various poultry species. Specifically, we observed that PLA2G12A underwent positive selection uniquely in dwarf guinea fowl, exhibiting significantly higher expression levels in quail breast muscles compared to the two domestic chicken breeds. This pattern closely correlated with variations in the content of a specific breast muscle lipid molecule, PC (15:0/18:2) (Pearson = −0.99968). Increased consumption of PC (15:0/18:2), a distinct component found in poultry meat, has been associated with a reduced risk of type 2 diabetes mellitus (T2D) in comparison to other meat sources [71]. Furthermore, our analysis revealed that the gene LPCAT2, which is involved in the conversion of LPC lipids to PC lipids in the breast muscles of domestic chicken breeds, experienced strong positive selection exclusively in AA broilers. The expression level of LPCAT2 in AA broiler breast muscles exceeded that in dwarf guinea fowl by more than threefold. This highlights the critical role of the LPCAT2 gene in governing the transformation of LPC-like lipids into PC-like lipids in the breast muscles of domestic chickens, with its expression significantly influenced by artificial directional selection. Moreover, the gene PTDSS1, known for encoding enzymes related to phospholipid biosynthesis [72], was the only gene enriched in the glycerophospholipid metabolism pathway of AA broilers, dwarf guinea fowl, and quail directly involved in regulating the breakdown of PC-like differentiated lipid molecules in the breast muscles. While there was no positive selection pressure detected on PTDSS1 in these avian species, its expression in breast muscles varied significantly among them, with quail exhibiting the highest levels, followed by AA broiler chicken and dwarf guinea fowl. This expression pattern was consistent with changes in the expression of PC-like differential lipids in the breast muscles of these species, suggesting that the PTDSS1 gene may play a pivotal role in influencing the deposition of PC-like differential lipids in the breast muscles of AA broilers, dwarf guinea fowl, and quail. Notably, the glycerophospholipid metabolic pathway has been associated with chicken flavor [73]. The distinct expression patterns of the aforementioned genes associated with this pathway among avian species under the influence of long-term artificial targeted selection underscore their significance as potential target genes for elucidating differences in meat flavor between fast-growing broilers, slow-growing local chickens, and other avian species.

The DGAT2 gene is recognized for catalyzing the final stage of triacylglycerol (TG) synthesis [74] and holds a pivotal role in energy balance, lipid storage, and intracellular lipid equilibrium [75]. Research has highlighted that DGAT2 overexpression boosts the expression of genes linked to lipid accumulation and adipogenesis, consequently elevating triacylglycerol levels within cells [76]. Conversely, reducing DGAT2 expression impedes SREBP-1 cleavage and fatty acid synthesis, as well as hepatic TG accumulation and release. In our investigation, we observed significant variations in the expression levels of the DGAT2 gene in the breast muscles of the four avian species, with pigeons exhibiting the highest levels, followed by quail and dwarf guinea fowl, while AA broilers displayed the lowest expression. Notably, DGAT2 underwent positive selection exclusively in AA broilers. This indicates that the expression of DGAT2 in avian breast muscles decreased progressively with the intensification of artificial directional selection. Consequently, the TG lipid content in the breast muscles of these species mirrored the expression pattern of the DGAT2 gene, with AA broiler breast muscles containing the lowest TG content. Furthermore, we identified strong correlations between the DGAT2 gene and specific differential lipid molecules, such as TG(16:1/18:2/20:4) (Pearson = 0.98682) and TG(18:2/20:4/20:4) (Pearson = 0.95868), underscoring the potential regulatory role of DGAT2 in TG-like lipid metabolism within avian breast muscles. However, its expression is notably influenced by the degree of artificial targeted selection. Given that TG content is crucial for intramuscular fat (IMF) deposition [77], with increased IMF enhancing muscle tenderness and flavor quality [78], DGAT2 can serve as a valuable marker gene for discerning meat quality discrepancies among birds subjected to varying degrees of selection pressure.

Domestic pigeons are taxonomically classified in a distinct order from domestic chickens and quails, indicating a considerable evolutionary separation. Their metabolic pathways for breast phosphatidylcholine (PC) and triacylglycerol (TG) differ significantly from those of the other three avian species and did not exhibit the previously discussed patterns of change or positive selective pressures. In particular, within the glycerophospholipid metabolism pathway, our study revealed that in pigeons, the PTDSS1 gene lost its function in PC breakdown, while the PEMT gene assumed a crucial role. The PEMT gene encodes an enzyme responsible for converting phosphatidylethanolamine to phosphatidylcholine in the liver. Activation of this gene leads to the synthesis of arachidonic acid (ARA), docosahexaenoic acid (DHA), and other PC-like lipids [79], enriching the pool of polyunsaturated fatty acids (PUFAs) crucial for organismal development [80]. Studies have shown a positive correlation between the PEMT gene and glycerophospholipid molecules like ARA and DHA in chicken breast muscle and abdominal fat [81]. Reduced expression of PEMT can result in diminished endogenous PC production in chickens, potentially predisposing them to hepatic steatosis [82]. Expression analysis demonstrated significant differential expression of the PEMT gene in the breast muscles of the four avian species, with pigeons exhibiting the highest expression levels, followed by quails, and the lowest levels observed in AA broilers. These findings highlight the pivotal role of PEMT in regulating PC-like lipid metabolism in pigeon breast muscles, shedding light on the molecular mechanisms behind the abundant presence of polyunsaturated fatty acids such as ARA and DHA in pigeon muscle tissue.

## 5. Conclusions

In this study, we conducted a comparative analysis of the lipidomics and transcriptomics of breast muscle tissue of AA broilers, dwarf guifei chickens, quails, and pigeons. The aim was to reveal subtle differences in their breast muscle lipid characteristics and to uncover the genetic basis that influences the development of these characteristics in breast muscles in these species under artificial selection pressure. Our findings highlight that the main lipids responsible for the differences in breast muscle mass traits among the four species of birds are phosphatidylcholine (PC) and triglycerides (TG). The key regulatory pathways involved were glycerophospholipid metabolism and glycolipid metabolism, and the key genes were identified as PLA2G12A, LPCAT2, DGAT2, and PEMT. The results of this study provide a valuable basis for further research on the nutritional characteristics and lipid composition of muscle tissue of different birds. But the function of key genes remains unclear, limited by the size of the sample and the population. Further studies are needed to verify the roles and mechanisms of these identified key genes, particularly those that are subject to positive selection during the formation of various breast muscle lipid features in birds, from an experimental perspective, especially through gene editing techniques.

## Figures and Tables

**Figure 1 animals-15-00694-f001:**
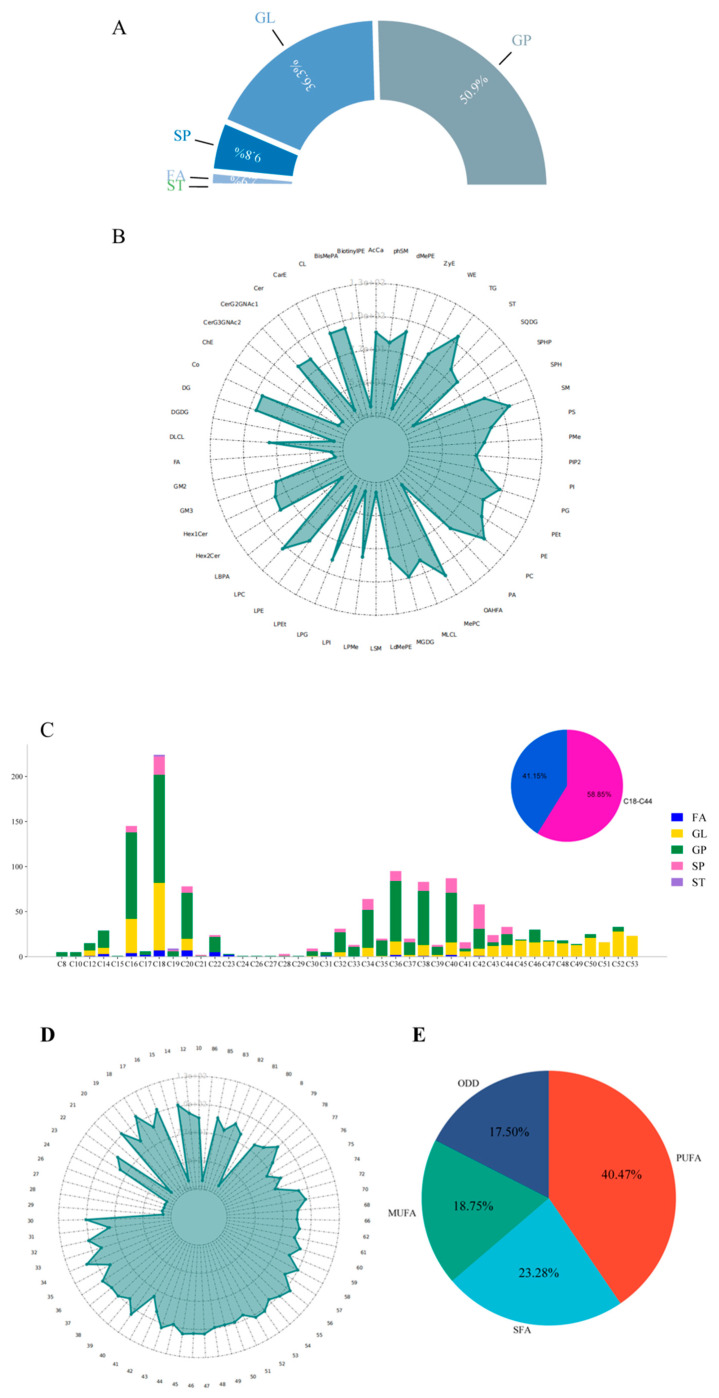
Characteristics of lipid group in poultry muscle of four kinds of poultry. Different percentages are shown in (**A**). (**B**) Radar map of the distribution of four poultry lipid subclasses: the outer circle represents the identified lipid subclass and the length of the green arrow in the inner circle represents the number of identified lipid subclasses. (**C**) Carbon chain length distribution diagram: the length of the histogram represents the number of lipid molecules under the carbon atom, and different colors represent different types of lipid molecules. The red in the upper right corner represents the proportion of C18-C44 carbon chain length of lipid molecules, and the blue represents the remaining carbon chain length of lipid molecules identified. (**D**) Carbon chain length distribution radar diagram: the outer circle represents the carbon chain length of the identified lipid molecule, the length of the green arrow in the inner circle represents the number of lipid molecules with the chain length identified. (**E**) Saturation pie chart: Different colors represent the percentage of saturated lipid molecules.

**Figure 2 animals-15-00694-f002:**
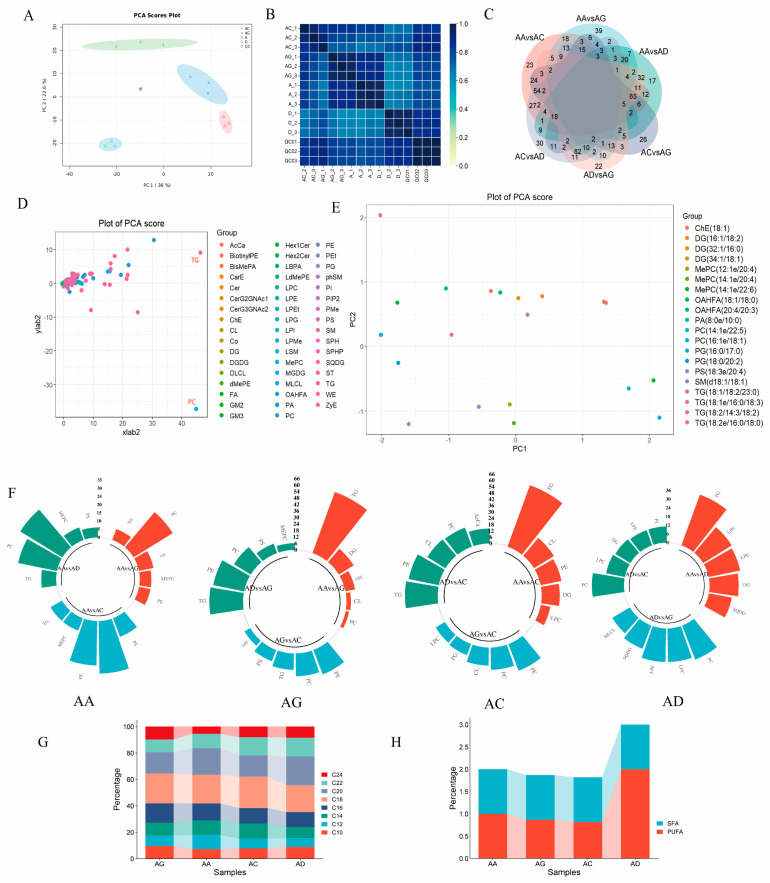
Comparative analysis of lipid groups in the breast muscles of four kinds of poultry. (**A**,**B**) Chest PCA heat map between three bioreplicates in each group of four kinds of poultry. The stronger the blue color, the stronger the correlation between the samples. (**C**) Differential lipid venn diagram. The four birds were divided into six comparison groups, and the petal center represented the common differential genes of the four birds. (**D**) Differential lipid PCA score map. The greater the distance of lipid representatives from the population, the higher the contribution to the difference in lipid distribution in the breast muscle of the four kinds of birds. (**E**) Lipid heat maps of the top 20 PCA scores in the breast muscles of four kinds of poultry. PCA calculation of the top ten lipid molecules showed that there was a high degree of separation between lipid molecules. (**F**) Four kinds of poultry breast muscle dominant lipid map. The four kinds of birds were divided into three groups, and the proportion of lipid content was indicated by the length of the column. (**G**) Dominant lipid carbon chain length. The height of the column represented the number of lipid molecules identified in the breast muscle of the bird; different colors represented the proportion of lipid molecules with different carbon chain lengths. (**H**) Dominant lipid unsaturated composition stack diagram. The height of the column represents the total lipid number of SFA and PUFA in birds; blue represents the proportion of SFA, and red represents the proportion of PUFA.

**Figure 3 animals-15-00694-f003:**
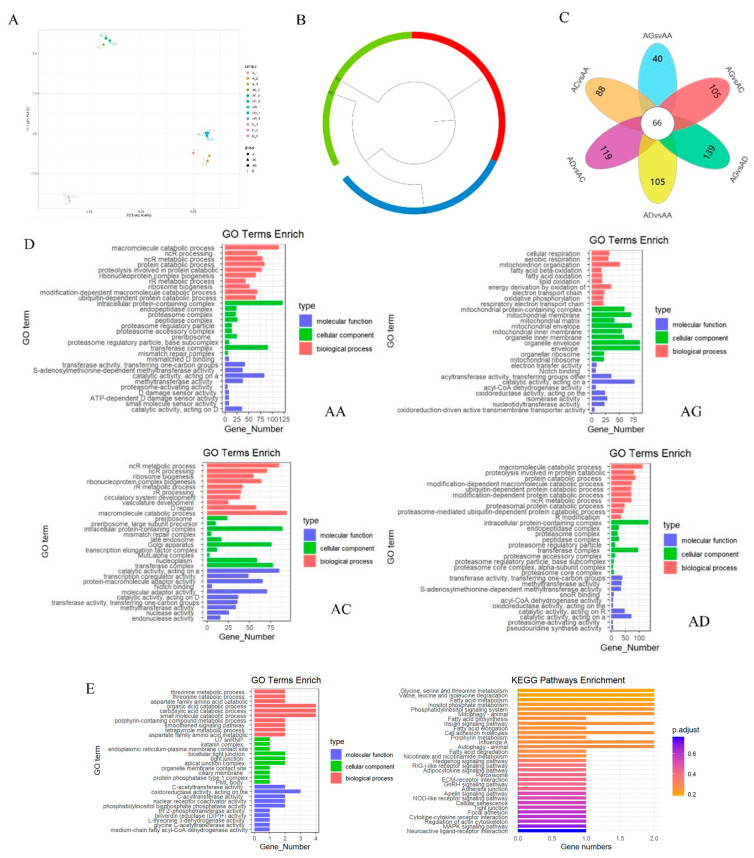
Transcriptome analysis of poultry muscle of four poultry species. (**A**) PCA heat map. Different colors represent different breeds of poultry breast muscle samples; the number of dots represents the number of samples and the farther the distance, the higher the degree of separation between samples. (**B**) Circular evolutionary tree. Different colors represent different birds; the greater the distance, the greater the evolutionary distance. (**C**) Difference orthologous genes Venn diagram. Different colors represent the number of differentially homologous genes between different groups and petal centers represent the common differentially homologous genes between the four birds. (**D**) Enrichment analysis of dominant lineal homologous gene GO. (**E**) Orthologous genes expressed in four kinds of common difference between birds GO and KEGG enrichment analysis results. The left side represents the GO enrichment results and the right side represents the KEGG enrichment results.

**Figure 4 animals-15-00694-f004:**
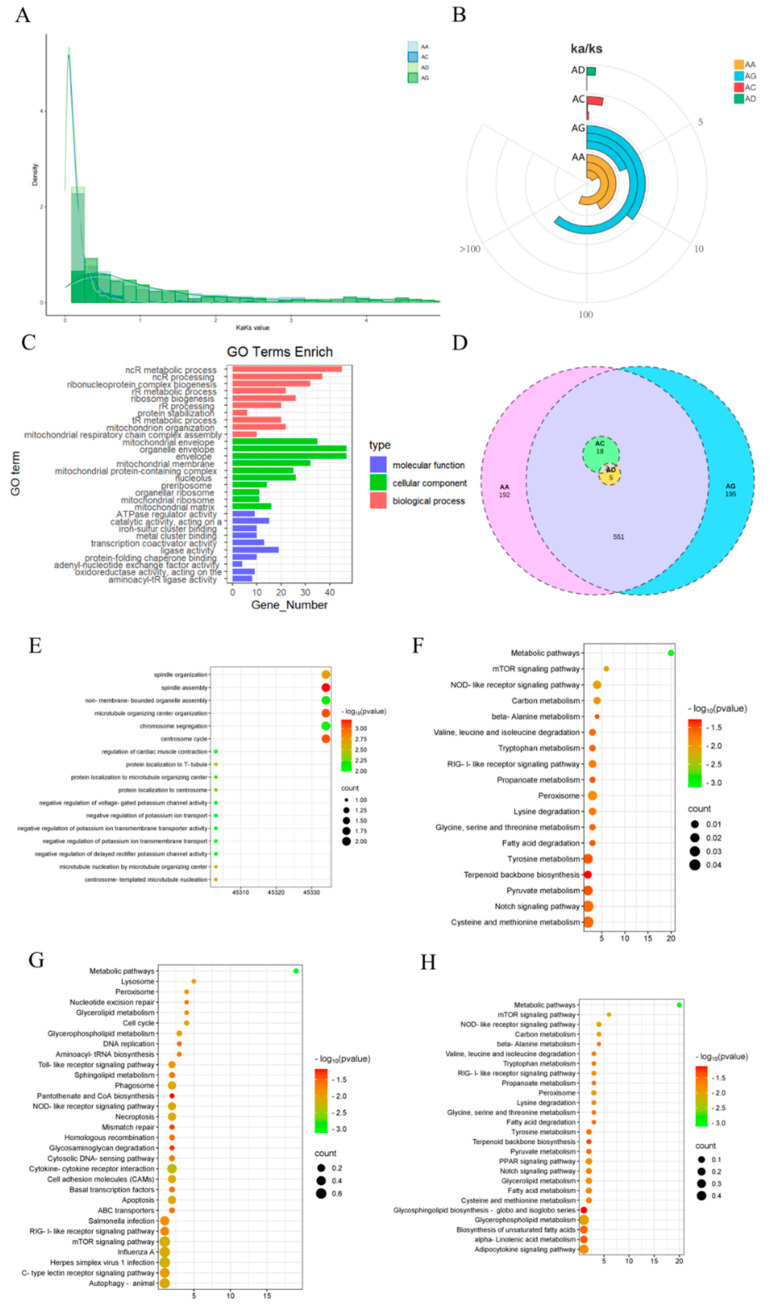
Analysis of adaptive evolution of homologous genes in poultry muscle of four poultry species. (**A**) Map of gene density distribution in the range 0 < *Ka/ks* < 5. The bar height represents the number of gene contributions under that stress. (**B**) *Ka/ks* distribution annular histogram. The column length represents the selective pressure distribution of the poultry. (**C**) Forward selection gene GO enrichment analysis histogram. (**D**) Venn map of selected genes. The intersection of circles represents the common gene, yellow and blue represent AC and AD, respectively, and the selected gene is contained within the common gene. (**E**) The GO enrichment results of the positive selection genes were shared by domestic chickens with quail or pigeons. (**F**) AA broilers and dwarf chicken wing common KEGG enrichment results by positive selection genes. (**G**) Only affected by KEGG enrichment of positive selection genes in AA broilers. (**H**) Only affected by KEGG enrichment of positive selection genes in Pygmy chicks.

**Figure 5 animals-15-00694-f005:**
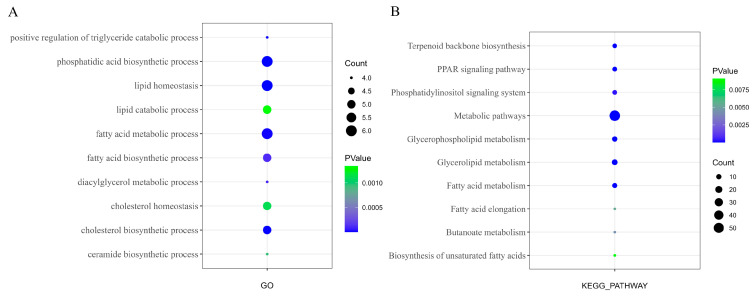
Functional enrichment analysis of positive selection genes related to lipid metabolism. (**A**) Positive selection for GO enrichment of genes related to lipid metabolism. (**B**) KEGG enrichment results of positive selection genes related to lipid metabolism with significant differential expression.

**Figure 6 animals-15-00694-f006:**
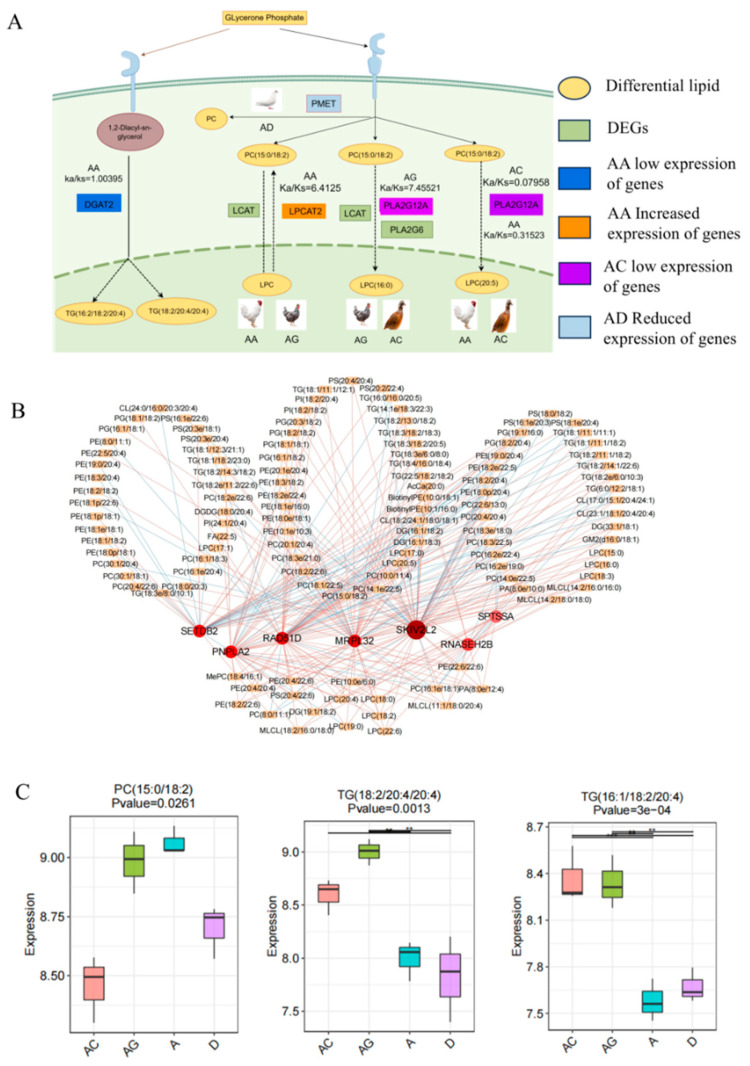
Correlation analysis of lipidome and transcriptome in four kinds of poultry breast muscle. (**A**) Comparison of characteristics of glycerophospholipid metabolism and glycerlipid metabolism in four kinds of poultry. (**B**) Positive selection of differentially expressed genes related to lipid metabolism and differential lipid molecular correlation network diagram. The red dot represents the core gene, the yellow dot represents the relevant lipid, and the thickness of the line represents the strength of the correlation (**C**) Comparison of the expression levels of differential lipid TG (16:1/18:2/20:4), TG (18:2/20:4/20:4), and PC (15:0/18:2) in the breast muscles of four kinds of poultry. The height of the column represents the expression level of the lipid molecule in the breast muscle of different birds.

## Data Availability

The authors state that data supporting the results of this study are available in the article and its Appendix A. All data are publicly available and, if necessary, can be obtained from the newsletter through reasonably request.

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
