# Peer review of "Multi-Omics Profiling of Lipid Variation and Regulatory Mechanisms in Poultry Breast Muscles"

_animals, 2025, doi:10.3390/ani15050694_

Round 1
Reviewer 1 Report
Comments and Suggestions for Authors
The manuscript presents a comprehensive lipidomic and transcriptomic analysis of pectoral muscles across four avian species, offering new insights into lipid composition differences and their molecular regulation. While well-structured and valuable for poultry meat quality research, certain areas require clarification and improvement, including methodology details, statistical analysis, discussion coherence, and overall readability. Additionally, the manuscript contains a large amount of information and is excessively long, making it essential to improve result presentation, emphasize key findings, and draw clearer conclusions.
To enhance accuracy and clarity, scientific and family names should be italicized, abbreviations should be defined upon first mention, references should be cited where needed, and typographical errors should be corrected. Furthermore, the methods section should be more specific and detailed to ensure reproducibility.
Simple Summary: The simple summary should be included as per the journal’s guidelines.
Abstract: The abstract is informative but too dense. Consider shortening it to improve readability while retaining key findings. Provide more details on the methods.
Line 18-20: Clearly define "AA" and specify the species and age.
Line 26: It would be helpful to define abbreviations like TG, PC, PE, Cer, and DG within the abstract.
Line 30-32: Clearly emphasize how the glycerophospholipid and glycerolipid pathways affect meat quality traits.
Introduction: The introduction is too long and should be more concise, focusing on the research gap and novelty. Detailed discussions on research and findings should be moved to the discussion section. The text covers multiple topics, including poultry production, lipidomics, transcriptomics, and genetic regulation—organizing these into clearer sections would improve readability.
While the study gap is well-defined, the research objectives should be stated more clearly. Additionally, including references to previous comparative lipidomic studies in poultry would strengthen the background. The authors mention the lack of lipid trait characterization in quail and pigeon meat but should explain why this gap exists to better justify the study. Since previous research has primarily focused on chicken phenotypic traits without transcriptomic and lipidomic data, clarifying how this study addresses that gap would provide better context.
Line 87-90: The taxonomic classifications in this section are incorrect. Chickens and quails belong to the Phasianidae family, with chickens in the Gallus genus and quails in the Coturnix genus. Pigeons belong to the Columbidae family and are classified under the Columba genus. The current names used in the text, such as Procapra for chickens and "Ornithoptera Doves" for pigeons, are incorrect and should be revised. Please verify.
Materials and Methods: Specific details for the methods need to be provided, including reagent sources, software versions, and parameter settings for bioinformatics tools.
Line 141-155: The animal selection criteria and sampling methods are clear, but more details on feed composition and rearing conditions would be helpful. Additionally, provide information on sample size, how the marketable age was determined, and the conditions in which the animals were kept before procurement.
Line 156-162: Briefly describe the RNA extraction process.
Line 164-175: Define MTBE and EP and specify the sources of the reagents.
Line 177-184: Include more details about the UHPLC-MS setup, such as the equipment model, mobile phase composition, gradient settings, column type, and other key conditions to make the method clearer and easier to replicate.
Line 183-185: Specify which database was used for lipid classification.
Line 187-195: Add details on how chain length and unsaturation were calculated. Also, explain how lipids were grouped into different classes and subclasses based on their structure.
Line 198-201: The criteria for identifying significant differential lipids (VIP > 1 and p < 0.05) make sense, but it would be helpful to explain why these cutoffs were chosen.
Line 203-212: Specify the RNA extraction kit used and any quality control steps taken.
Line 214-232: Provide clear filtering thresholds for low-quality reads, N rate, and sequence length in fastp to explain the quality control process.
Lines 234-239: Include details on the inflation value or threshold settings used in OrthoFinder and the Markov clustering algorithm (MCL) for better clarity.
Lines 240-251: Mention the normalization method used in RSEM for TPM calculation, such as FPKM normalization or batch correction, to ensure consistency.
Lines 252-260: Specify the exact model used in KaKs calculator (e.g., NG, ML, LWL) to accurately determine selection pressure.
Lines 261-274: Explain why O2PLS was chosen for correlation analysis instead of other methods like WGCNA or CCA to justify the approach.
Statistical Analysis: Include more details on sample size, statistical tests performed, and the number of replicates for each study parameter to improve reliability.
Results: Needs stronger emphasis on statistical analysis.
Sections 3.1 & 3.2: These could be combined and improved by directly comparing lipid profiles across avian species rather than discussing overall trends in 3.1 and differences in 3.2 separately.
Line 277-310: Clarify if there was any variation in lipid classes between samples, as standard deviation is not mentioned.
Line 311-322: The PCA results are explained, but variance values should be provided to confirm the appropriateness of the selected components.
Line 356-395: Support gene expression comparisons with hierarchical clustering or heatmaps to better illustrate inter-species variations.
Line 401-402: Justify why Ka/Ks = 999 or 0.0001 was used as an exclusion criterion—was this based on biological or technical factors?
Line 404-406: The Ka/Ks distribution across species is insightful, but statistical significance should be explicitly stated. Were statistical tests used to compare selection pressures?
Line 414-417: The mention of positive selection genes linked to protein synthesis, lipid metabolism, and immunity is too broad—include specific key genes.
Line 475: Specify which statistical test was used to determine significant differential expression. Were FDR corrections applied?
Line 493: The term "significantly enriched" is vague—provide p-values or FDR-adjusted values to support significance claims.
Line 510-520: The PLA2G12A gene shows different selection pressures—were these differences statistically significant?
Line 522-526: The DGAT2 gene has different selection pressures across species—does this correlate with lipid deposition differences? Consider expanding on this.
Line 540-552: The gene-lipid correlation network is useful, but add a statistical measure (e.g., Pearson/Spearman correlation values) to strengthen the claims.
Figures 1-4: Figures lack clarity—labels should be more visible and distinct, especially for lipid subclasses and metabolic pathways.
Figure & Table Legends: Some are missing full descriptions of abbreviations and explanations. They should be standalone so they can be easily understood without needing to refer to the manuscript.
Discussion
Line 571-572: Clarify if the 1,542 lipid molecules across 50 subclasses were the same in all species or if they varied.
Line 576-577: Include references to past studies on chicken, pigeon, and quail lipid composition for stronger scientific support.
Line 581-583: Explain how PC, PE, and TG contribute to meat quality traits like tenderness and juiciness.
Line 597-598: The higher SFA levels in broiler pigeons are interesting—do they relate to metabolic rate or flight adaptations?
Line 600-606: Strengthen the evolutionary discussion by adding references to genome studies on poultry lipid metabolism.
Line 609-613: Explain what genetic factors might cause quail and dwarf guinea fowl to have similar lipid profiles.
Line 615-621: Provide data on selection intensity for lipid traits in broilers to support claims about artificial selection pressure.
Line 634-636: Mention which statistical tests were used to confirm significance in the 711 differentially expressed lipid molecules.
Line 642-647: Add numerical values comparing TG and PC levels across species for a stronger discussion.
Line 654-662: Expand on how glycerophospholipid and glycerolipid metabolism impact lipid transport and oxidation.
Line 669-671: Explain how PC storage of PUFAs affects meat texture and oxidative stability.
Line 676-678: Were PLA2G12A expression levels significantly different across species? Provide statistical values.
Line 689-692: Discuss if positive selection of LPCAT2 in AA broilers is linked to their faster growth rates.
Line 696-701: Were the regulatory pathways of PTDSS1 analyzed? Explain its role in lipid deposition.
Line 712-717: Since DGAT2 is important for TG synthesis, were other DGAT family genes (e.g., DGAT1) studied?
Line 719-722: The inverse correlation between DGAT2 expression and selection intensity is important—compare it to other livestock species.
Line 726-728: Add statistical validation (e.g., Pearson correlation, p-values) for DGAT2-TG correlations.
Line 739-741: The loss of PTDSS1 function in pigeons—was this due to a mutation or gene regulation change?
Line 745-747: Provide PUFA concentration data in pigeons to support PEMT’s role in phosphatidylcholine metabolism.
Line 749-751: Discuss the benefits of higher PEMT expression in pigeons.
Line 759-761: Include data showing that PC and TG are the main lipids influencing meat quality.
Line 764-766: Explain how key genes (PLA2G12A, LPCAT2, DGAT2, PEMT) can be used in poultry breeding programs.
Line 768-770: Suggest future studies to confirm the roles of these genes using functional genomics or gene-editing experiments.
Author Response
Comments and Suggestions for Authors
Simple Summary: The simple summary should be included as per the journal’s guidelines.
Author response:Thank you for your suggestions. We regret that we previously overlooked the brief summary of the article. I sincerely apologize for any inconvenience this oversight may have caused. The relevant content has now been added to the revised manuscript.
Abstract: Considering your suggestion in the abstract to potentially shorten the content while retaining key results and enhancing readability, and given that Lines 18-20, Line 26, and Lines 30-32 are all related to the abstract, along with similar feedback from other reviewers, we have rewritten the abstract to address these concerns.
Author response: Thank you for the reminder. Considering that other reviewers have put forward similar opinions to you on the abstract, I have further optimized this part of the content on the basis of the original, and modified part of the content to highlight the objectives, methods, key results and impacts as much as possible without changing the meaning.
Introduction:
Author response: Thank you for your detailed and specific suggestions regarding the Introduction section. We have rewritten the content of this section to better organize the information and improve clarity and readability. Additionally, as you pointed out, we have revised the information related to the taxonomic classification of chickens, quail, and pigeons during the rewriting process.
Line 141-155: The animal selection criteria and sampling methods are clear, but more details on feed composition and rearing conditions would be helpful. Additionally, provide information on sample size, how the marketable age was determined, and the conditions in which the animals were kept before procurement.
Author response: Thank you very much for your suggestions.Your suggestion was very helpful, so I have rewritten this section to include information on the sample size, the method used to determine the age at market, and the conditions under which the animals were kept prior to purchase.
Line 156-162: Briefly describe the RNA extraction process.
Author response: Thank you very much for your suggestions. Your suggestions are very helpful for us to improve this article, so we have rewritten this part of the content
Line 164-175: Define MTBE and EP and specify the sources of the reagents.
Author response: Thank you very much for your suggestions. In response to your question about the source of reagents, we have re-labeled them in article Line 164-175
Line 177-184: Include more details about the UHPLC-MS setup, such as the equipment model, mobile phase composition, gradient settings, column type, and other key conditions to make the method clearer and easier to replicate.
Author response: Thank you very much for your suggestions. Thanks for your comments, we have checked and supplemented all the equipment in Line 177-184 as well as the key conditions of the test.
Line 183-185: Specify which database was used for lipid classification.
Line 187-195: Add details on how chain length and unsaturation were calculated. Also, explain how lipids were grouped into different classes and subclasses based on their structure.
Author response: Thank you very much for your suggestions. I intend to provide a unified answer to your questions on lipid classification and database sources raised in Line 183-185 and Line 187-195. The classification of lipids has always been a complex and difficult matter. There have been many ways to classify lipids throughout history, but classifications based on structure and synthetic pathways are the most accurate. Our current approach to classifying lipids by structure comes from the Lipid Metabolites and Pathways Strategy, which was funded by the National Institutes of Health (NIH) in 2003. Lipid classification system Lipid classification system Lipid texture map proposed.
Line 198-201: The criteria for identifying significant differential lipids (VIP > 1 and p < 0.05) make sense, but it would be helpful to explain why these cutoffs were chosen.
Author response: Thank you very much for your suggestions. In structural equation model (SEM), detecting the importance index (VIP) and significance level (P-value) of variables is an important step in the evaluation model construction. Selecting the criteria of VIP > 1 and p < 0.05 can ensure the stability of the model and the validity of the variables.First, the VIP index measures how well the underlying variable explains the dependent variable. When the VIP value is greater than 1, it indicates that the latent variable plays a significant role in the model, usually excluding the negative impact of the variable on the model's explanatory ability, which is conducive to the stability and interpretability of the model. This criterion refers to the study by Bido et al. (2017), who suggested VIP > 1 as the criterion for determining the importance of variables.Second, the significance level of p < 0.05 controls the probability of Class I errors, that is, errors that reject the correct model hypothesis. This criterion helps ensure the significance of variables and avoids false conclusions due to chance. Henseler et al. (2012) emphasized the importance of using P-value for significance test when discussing PLS-SEM method.In summary, the combination of VIP > 1 and p < 0.05 provides an effective screening criterion for the model and ensures the accuracy and reliability of the construction. This is the relevant literature
Bido, D., Fichet, B., & LEBLANC, F. (2017). Variable importance in partial least squares structural equation modeling (VIP) using XLstat-PLS. Journal of Applied Statistics, 44(11), 2017.
Henseler, J., Ringle, C. M., & Sarstedt, M. (2012). How to reggressially assess the relevance of constructs in PLS-SEM: A comparison of different methods. Business & Information Systems Engineering, 55(4), 177-188.
Hair, J. F., Hult, G. T., Ringle, C. M., & Sarstedt, M. (2014). A primer on partial least squares structural equation modeling (PLS-SEM). 2nd ed.
Line 203-212: Specify the RNA extraction kit used and any quality control steps taken.
Line 214-232: Provide clear filtering thresholds for low-quality reads, N rate, and sequence length in fastp to explain the quality control process.
Lines 234-239: Include details on the inflation value or threshold settings used in OrthoFinder and the Markov clustering algorithm (MCL) for better clarity.
Lines 240-251: Mention the normalization method used in RSEM for TPM calculation, such as FPKM normalization or batch correction, to ensure consistency.
Lines 252-260: Specify the exact model used in KaKs calculator (e.g., NG, ML, LWL) to accurately determine selection pressure.
Author response: Thank you very much for your suggestions. Considering that what you mentioned in this part is all about further improving the article to enhance the accuracy of the article, I hereby give a unified answer to these suggestions, and I have supplemented and improved these things in the article.
Lines 261-274: why O2PLS was chosen for correlation analysis instead of other methods like WGCNA or CCA to justify the approach.
O2PLS (Two-Way Orthogonal PLS) means bidirectional orthogonal partial least squares. Compared with PCA, PLS WGCNAand CCA, this method takes into account the size, scale, distribution and experimental error of data sets in different scenarios, and considers the joint, specific and residual parts between different data sets in the modeling process, which is suitable for data mining in complex scenarios. It is a type of unsupervised modeling.
Conclusion
Line 277-310: Clarify if there was any variation in lipid classes between samples, as standard deviation is not mentioned.
Author response: Thank you very much for your suggestions. In this part, we did not mention whether there were differences in lipid categories among samples because we statistically analyzed the clustering among samples through PCA values and conducted comprehensive analysis on the three samples of the same variety after meeting the reliability. This part is explained between Line333-335.
Line 311-322: The PCA results are explained, but variance values should be provided to confirm the appropriateness of the selected components.
Line 356-395: Support gene expression comparisons with hierarchical clustering or heatmaps to better illustrate inter-species variations.
Author response: Thank you very much for your suggestions.Your suggestions are useful for further visualizing the differences between the four types of bird breast muscles, but we have taken this section as a table to summarize, focusing on the functional differences of genes.
Line 401-402: Justify why Ka/Ks = 999 or 0.0001 was used as an exclusion criterion—was this based on biological or technical factors?
Line 404-406: The Ka/Ks distribution across species is insightful, but statistical significance should be explicitly stated. Were statistical tests used to compare selection pressures?
Author response: Thank you very much for your suggestions. I intend to give a unified answer to the questions you mentioned in Line 401-402 and Line 404-406.The exclusion criteria for extreme Ka/Ks values (such as 999 or 0.0001) are mainly based on technical factors, and are affected by when the synonymous replacement rate (Ks) is extremely low or even approaching 0 and when the non-synonymous replacement rate (Ka) approaches 0. In order to ensure the accuracy of analysis results, We chose Ka/Ks = 999 or 0.0001 as the exclusion criteria. As for your idea of using statistical tests to compare selection pressure, According to Navarro A., & Barton, N. H. (2003). Chromosomal speciation and molecular divergence -- accelerated evolution in rearranged chromosomes. Science, 300(5617), 321–324. The idea of this paper is analyzed, and the method of statistical test is not used in the comparison of selection pressure.
Line 414-417: The mention of positive selection genes linked to protein synthesis, lipid metabolism, and immunity is too broad—include specific key genes.
Author response: Thank you very much for your suggestions. In response to your questions, I have re-written the paragraph of Line 414-417.
Line 475: Specify which statistical test was used to determine significant differential expression. Were FDR corrections applied?
Line 493: The term "significantly enriched" is vague—provide p-values or FDR-adjusted values to support significance claims.
Author response: Thank you very much for your suggestions. Here, I will focus on the answers to these two questions. All the genes screened in this study were FDR corrected, which satisfied log2|FC| >= 1 and FDR<0.05.
Line 510-520: The PLA2G12A gene shows different selection pressures—were these differences statistically significant?
Author response: Thank you very much for your suggestions. In our study, we demonstrated that the PLA2G12A gene exhibited distinct selection pressures under different conditions. This data originates from our aforementioned analysis of selection pressures across homologous genes, with detailed enumeration of the selection pressures acting on PLA2G12A and other relevant genes in bird muscle tissue provided in Supplementary Material 4.
Line 522-526: The DGAT2 gene has different selection pressures across species—does this correlate with lipid deposition differences? Consider expanding on this.
Author response: Thank you very much for your suggestions. In response to your question, although we found a strong consistency between the distribution of TG in these four kinds of birds and the expression of DGAT2 gene and selection pressure, considering the different genetic basis among different birds, we cannot guarantee that the selection pressure of DGAT2 is the key factor leading to the difference in lipid deposition. Therefore, we describe the observed data in Line 522-526.
Line 540-552: The gene-lipid correlation network is useful, but add a statistical measure (e.g., Pearson/Spearman correlation values) to strengthen the claims.
Author response: Thank you very much for your suggestions. For this part of relevant data, I have provided corresponding data in Supplementary data 8, and mentioned this supplementary material in Line547-550. In addition, the correlation coefficient of this part of important genes and its related lipid molecule Person has also been described.
Figures 1-4: Figures lack clarity—labels should be more visible and distinct, especially for lipid subclasses and metabolic pathways.
Figure & Table Legends: Some are missing full descriptions of abbreviations and explanations. They should be standalone so they can be easily understood without needing to refer to the manuscript.
Author response: Thank you very much for your suggestions.Considering that other reviewers also raised the same questions about this part, I redraw the picture, re-wrote the text description, and modified the supplementary materials.
Dicussion
Line 571-572: Clarify if the 1,542 lipid molecules across 50 subclasses were the same in all species or if they varied.
Author response: Thank you very much for your suggestions. These 1542 lipid molecules are the sum of all the lipid molecules identified in the four types of poultry breast muscle, and considering that the starting point of this study was to explore the genetic basis of the differences in lipid metabolism of these four types of poultry, So we put together all the different lipids in the four types of poultry breast muscle for discussion and we refer to Supplementary data 1 in Line 275-277 for details on the specific classification.
Line 576-577: Include references to past studies on chicken, pigeon, and quail lipid composition for stronger scientific support.
Author response: Thank you very much for your suggestions. The references in this paragraph provide references to relevant research.
Line 581-583: Explain how PC, PE, and TG contribute to meat quality traits like tenderness and juiciness.
Author response: Considering that other reviewers also raised the same questions about this part, we re-wrote this part, focusing on the influence of TG, PC and PE lipid molecule content on meat quality.
Line 597-598: The higher SFA levels in broiler pigeons are interesting—do they relate to metabolic rate or flight adaptations?
Author response: Thank you very much for your suggestions. The ratio of PUFA to SFA is closely related to the health of meat quality, which is influenced by multiple factors such as genetics and feeding environment. Your question raises a very new perspective, and there are no studies to prove that it is related to metabolic rate or flight adaptation.
Line 600-606: Strengthen the evolutionary discussion by adding references to genome studies on poultry lipid metabolism.
Line 609-613: Explain what genetic factors might cause quail and dwarf guinea fowl to have similar lipid profiles.
Author response: Thank you very much for your suggestions. The question you raised makes sense to us, but at present, limited by the scale of genomic studies on lipid metabolism in poultry and the genetic factors behind the lipid similarity between quail and Taffy chickens, we have not collected any possible explanations or data. At present, the only studies mainly focus on the lipid differences and pathways between the breast muscles of poultry. This is also the message that we want to further elaborate in this study.
Line 615-621: Provide data on selection intensity for lipid traits in broilers to support claims about artificial selection pressure.
Author response: Thank you very much for your suggestions. At present, there are still few researches on selection intensity of broiler traits, which mainly focus on growth rate and feed conversion rate, etc. I have expounded these data in the 56th reference, that is, lines 615-619 of the article. At present, we have not found any relevant articles published by our peers on the data related to lipid selection pressure.
Line 634-636: Mention which statistical tests were used to confirm significance in the 711 differentially expressed lipid molecules.
Author response: Thank you very much for your suggestions. All the differential lipid screens in this study met the industry-recognized threshold of VIP>1, p value<0.05, and the relevant thresholds were also mentioned in Line 634 after screening the 711 lipids.
Line 642-647: Add numerical values comparing TG and PC levels across species for a stronger discussion.
Author response: Thank you very much for your suggestions.TG and PC lipids belong to a large category and contain many specific molecules. The method we adopted in the analysis was to compare the four types of birds in pith and pith groups, and it was found that there were large differences in the number of small molecules of TG and PC lipids between different birds. Therefore, in order to ensure the effectiveness and accuracy of the paper, we did not directly describe the concentration value of specific molecular compounds. However, we have provided all the data in the table in supplement 1, In order to further ensure the accuracy of the article, we have added a description of supplementary materials on Line647.
Line 654-662: Expand on how glycerophospholipid and glycerolipid metabolism impact lipid transport and oxidation.
Author response: Thank you very much for your suggestions. We have supplemented the content of this part and further elaborated on the regulation of fats by glycerin and glycerophospholipids.
Line 669-671: Explain how PC storage of PUFAs affects meat texture and oxidative stability.
Author response: Thank you very much for your suggestions. We supplement this part in Line669-671.
Line 676-678: Were PLA2G12A expression levels significantly different across species? Provide statistical values.
Author response: Thank you very much for your suggestions. The expression level of PLA2G12A gene is obviously different among the four poultry species. This data has been mentioned in Supplementary 3. In order to ensure that the data in the article is too redundant, this data is not directly described in the main text.
Line 689-692: Discuss if positive selection of LPCAT2 in AA broilers is linked to their faster growth rates.
Author response: Thank you very much for your suggestions.The question you raised is very enlightening to us, but considering that, on the one hand, LPCAT2 gene is one of the marker genes of lipid metabolism, this study only focuses on the correlation between its selective pressure intensity and lipid metabolism; on the other hand, there is no relevant literature supporting the correlation between LPCAT2 gene selection pressure and its growth.
Line 696-701: Were the regulatory pathways of PTDSS1 analyzed? Explain its role in lipid deposition.
Author response: Thank you very much for your suggestions. Your suggestion is helpful to us, but in the process of analysis, we analyzed the regulatory mechanism of PTDSS1 gene. Considering that the description in this paragraph was mainly for pigeons, and PTDSS1 gene did not meet the strong analytical differences, we did not elaborate its regulatory mechanism in this description.
Line 712-717: Since DGAT2 is important for TG synthesis, were other DGAT family genes (e.g., DGAT1) studied?
Author response: Thank you very much for your suggestions. We have conducted relevant studies on all the genes in the DGAT family you mentioned, especially the DGAT1 gene you mentioned. Unfortunately, only DGAT2 gene in the DGAT family meets all the screening conditions, which may be related to its important role in TG synthesis.
Line 719-722: The inverse correlation between DGAT2 expression and selection intensity is important—compare it to other livestock species.
Author response: Thank you very much for your suggestions. The comparison of the relationship between selection intensity and expression of DGAT2 in a larger population as you mentioned is a valuable proposal. However, unfortunately, we calculated the relationship between selection pressure and expression amount based on homologous genes when we made this analysis. However, relevant studies on other species are still lacking at present, and we have tried to collect relevant data. Unfortunately, we don't have reliable data to use.
Line 726-728: Add statistical validation (e.g., Pearson correlation, p-values) for DGAT2-TG correlations.
Author response: Thank you very much for your suggestions. TG lipids are a large category containing many small molecules. In the description of the paragraph Line 726-728, we emphasized the correlation Pearson coefficient and considered P-values. In addition, we also demonstrated these data in detail in Supplement 8.
Line 739-741: The loss of PTDSS1 function in pigeons—was this due to a mutation or gene regulation change?
Author response: Thank you very much for your suggestions. For the loss of function of PTDSS1 gene in pigeons, we still lack data to explain the reasons behind this phenomenon. In the future, we also plan to explain the reasons of the loss of function of this gene by regulation or mutation.
Line 745-747: Provide PUFA concentration data in pigeons to support PEMT’s role in phosphatidylcholine metabolism.
Author response: Thank you very much for your suggestions. We have also considered your question, but PUFA is a general term for a large class of lipids, including various lipid molecules and designed biological metabolic processes. This part of data is not of great significance to support the role of PEMT in phosphatidylcholine, so we did not emphasize this part in the paper.
Line 749-751: Discuss the benefits of higher PEMT expression in pigeons.
Author response: Thank you very much for your suggestions.
We have tried to search for information. At present, PEMT is still rarely studied in birds, especially pigeons, especially in lipid metabolism. At present, there is only one paper on embryonic development, which has no obvious supporting significance for our research. Futterman MA, García AJ, Zamir EA. Evidence for partial epithelial-to-mesenchymal transition (pEMT) and recruitment of motile blastoderm edge cells during avian epiboly. Dev Dyn. 2011 Jun;240(6):1502-11. doi: 10.1002/dvdy.22607. Epub 2011 Mar 15. PMID: 21412939; PMCID: PMC3128786.
Line 759-761: Include data showing that PC and TG are the main lipids influencing meat quality.
Author response: Thank you very much for your suggestions. When writing the article, we tried to further describe the TG and PC data in this paragraph. However, considering that this part of data has been explained many times in the results part of the article and the discussion line 568-593, in order to avoid the verbose article caused by data redundancy, we did not contempt it again in this part.
Line 764-766: Explain how key genes (PLA2G12A, LPCAT2, DGAT2, PEMT) can be used in poultry breeding programs.
Line 768-770: Suggest future studies to confirm the roles of these genes using functional genomics or gene-editing experiments.
Author response: Thank you very much for your suggestions. Your suggestion is very helpful to us, so we have added a summary section based on your suggestion, and added a recommendation explaining how key genes are used in poultry breeding and using gene editing to determine the role of these genes.

Reviewer 2 Report
Comments and Suggestions for Authors
Manuscript animals-3495712, entitled “The integrated lipidomic and transcriptomic analysis unveiled variations in lipid composition and their associated regulatory mechanisms within the breast muscles of four distinct poultry species”
Recommendation: The above paper is not suitable for publication in its present form.
This article provides information on the variations in lipid composition and their associated regulatory mechanisms within the breast muscles of four distinct poultry species. It is in general appropriately organized, carried out and written, however there are some points that should be corrected or clarified.
Numbers in the references in text does not correspond to the reference list.
L279 and throughout the text: Supplementary data? Do you mean supplementary tables?
L15: “…are the major…”
L50: “…poultry in China by May 2020.”
L60-61: “However, a lack of detailed characterization and comparative studies on the meat quality attributes of these avian species remains.”
L73, 105, 115: “there is” instead of “there exists”
L104: “…a few selected domestic…”
L149: “They were reared with free access to feed and water…”
L150-151: Please check reference style
L152: D or AD?
L165-175: Please convert imperative to indicative grammatical mode
L253: “used” instead of “taken”
L259: “in accordance to” instead of “using”
L270: 1542? Where is this number shown?
L283-286: Different percentages are shown in Figure 1A
L302: C9 and C11 are not shown in Fig. 1C
L308: 30:1?
L315: 711? Where is this number shown?
L316: “Among them”
L363: 5790 or 5670?
L379: In Fig. 3D, the group AD is missing and two AG are presented
L499: These data are not shown in Table S6
Figure 6C: What about LPC?
L584: “The dominant lipid compositions”?
Comments on the Quality of English LanguageThe English could be improved to more clearly express the research
Author Response
Comments and Suggestions for Authors
1: I suggest shortening the title of the paper.
Author response: Thank you for your suggestions. Line279 and the entire text refer to the supplementary table, there is a problem with my description, which has been corrected.
2: L15: “…are the major…”
L50: “…poultry in China by May 2020.”
L60-61: “However, a lack of detailed characterization and comparative studies on the meat quality attributes of these avian species remains.”
L73, 105, 115: “there is” instead of “there exists”
L104: “…a few selected domestic…”
L149: “They were reared with free access to feed and water…”
Author response: Thank you for pointing out the errors in the content of these articles. I have checked and made changes
3: L150-151: Please check reference style
Author response: Thank you for your suggestions. Reference formatting errors have been corrected
4: L152: D or AD?
Author response: Thank you for your suggestions. This part refers to AD and the error has been corrected.
5: L165-175: Please convert imperative to indicative grammatical mode
Author response: Thank you for your suggestions. This section has been changed.
6: L253: “used” instead of “taken”、L259: “in accordance to” instead of “using”
Author response: Thank you for your suggestions. This section has been changed.
7:L270: 1542? Where is this number shown?
Author response: Thank you for your suggestions. This figure is for the total number of lipid molecules detected, which we did not directly reflect in the paper, but only mentioned in the attached table, in addition, we also mentioned this figure in the discussion section.
8: L283-286: Different percentages are shown in Figure 1A
Author response: Thank you for your suggestions. This section has been changed to: the different percentages of lipids identified in the pectoral muscles of the four species of birds.
9: L302: C9 and C11 are not shown in Fig. 1C
Author response: Thank you for your suggestions. We tested this data and found no correlation between the two short single-chain lipids, C9 and C11.
9: L302: C9 and C11 are not shown in Fig. 1C
Author response: Thank you for your suggestions.Yeah, we re-examined the data and ran the data, and we confirmed that it was 30:1.
10: L315: 711? Where is this number shown?
Author response: Thank you for your suggestions. In our study, 711 differential lipids were identified, which were not directly represented in the pictures, but were labeled in Table S1
11: L363: 5790 or 5670?
Author response: Thank you for your suggestions. I'm sorry, but due to our mistake, this section identified 5,610 homologous genes.
12: L379: In Fig. 3D, the group AD is missing and two AG are presented
Author response: Thank you for your suggestions. Sorry, due to our mistake, we have redrawn and uploaded these pictures.
13:L499: These data are not shown in Table S6
Author response: Thank you for your suggestions. This part is because our description is not accurate enough. The original intention of Table S6 is to describe the enrichment of gene pathways in the breast muscle of birds, so we deleted the inaccurate description.
14: Figure 6C: What about LPC?
Author response: Thank you for your suggestions.The purpose of this part of the chart is to explain the expression levels of lipids that are highly correlated with the target gene DGAT2 in the breast muscles of the four poultry species, so LPC is not explained.
15: L584: “The dominant lipid compositions”?
Author response: Thank you for your suggestions.This major lipid composition is inaccurate in our description and has been changed to "changes in the content of these lipid molecules have a large effect on meat quality."
16: Comments on the Quality of English Language
The English could be improved to more clearly express the research
Author response: Thank you for your suggestions.I am very sorry that this is my first submission and there are major problems in language description and expression. Other reviewers have also put forward the same suggestions. In the process of revision, I have tried my best to revise other problems in the article.

Reviewer 3 Report
Comments and Suggestions for Authors
The manuscript presents a comprehensive multi-omics analysis of lipid composition and gene regulation in the breast muscles of four avian species. The integration of lipidomics and transcriptomics is valuable and enhances understanding of poultry meat quality and genetic selection. However, some areas require clarifications in methodology, deeper discussion of findings, and minor textual improvements:
- The abstract provides a strong summary but should explicitly state the novelty of this study compared to previous lipidomic or transcriptomic studies in poultry.
- Including a key numerical result from the lipidomic or transcriptomic findings would strengthen the abstract.
- The introduction mentions that artificial selection affects lipid traits but lacks a clear explanation of how different selection pressures shape lipid composition. Briefly discuss previous studies on selection impacts in poultry breeds.
- The study selects three individuals per species. Given the potential biological variation, was this sample size sufficient for statistical power? Justify why this number is adequate or discuss limitations.
- The positive selection threshold (Ka/Ks > 1) is used, but genes with Ka/Ks > 100 are included. Are such extreme values biologically meaningful? Consider addressing potential artifacts in selection pressure estimates.
- The paper applies OPLS-DA for lipid screening. Were multicollinearity, normality, or variance assumptions tested before statistical modeling? If not, add a discussion on potential model limitations.
- The study finds DGAT2 under positive selection in broilers but with lower expression. Does this suggest that selection has led to reduced TG synthesis? A more detailed explanation is needed.
- The manuscript provides detailed lipid classification (TG, PC, LPC), but how do these lipid differences affect meat quality traits (e.g., tenderness, flavor)? Expanding on this would enhance practical relevance.
- The PCA results show separation between lipid classes, but the discussion does not address whether this aligns with genetic divergence among species. A brief mention would improve clarity.
- Some figures (e.g., PCA heat maps) lack sufficient explanation in captions. Adding a sentence on what the figure demonstrates would improve readability.
Author Response
Comments and Suggestions for Authors
1: The abstract provides a strong summary but should explicitly state the novelty of this study compared to previous lipidomic or transcriptomic studies in poultry.
2: Including a key numerical result from the lipidomic or transcriptomic findings would strengthen the abstract.
Author response: Thank you for your suggestions. Considering that other reviewers have raised similar suggestions regarding the abstract section, we have rewritten the abstract and incorporated key numerical data to strengthen its content. In light of the concise summary in the preceding paragraph and the subsequent elaboration in the abstract on the differences between this study and previous research, we have refrained from further adding this content in the abstract to avoid redundancy.
3: The introduction mentions that artificial selection affects lipid traits but lacks a clear explanation of how different selection pressures shape lipid composition. Briefly discuss previous studies on selection impacts in poultry breeds.
Author response: Thank you for your suggestions.
4:The study selects three individuals per species. Given the potential biological variation, was this sample size sufficient for statistical power? Justify why this number is adequate or discuss limitations.
Author response: Thank you for your suggestions. Your suggestions made us realize the shortcomings of our study. In biological research, three biological replicates can meet the research needs, but the expansion of sample size is very important to improve the accuracy of statistical results. Therefore, we discussed the shortcomings of using only three samples in the discussion section of Chapter 5 of this paper.
5:The positive selection threshold (Ka/Ks > 1) is used, but genes with Ka/Ks > 100 are included. Are such extreme values biologically meaningful? Consider addressing potential artifacts in selection pressure estimates.
Author response: Thank you for your suggestions. In response to your questions about the biological significance of positive selection thresholds (Ka/Ks > 1) and extremes (Ka/Ks > 100), we take your comments seriously and respond in detail here.
The Ka/Ks ratio is a commonly used index to estimate the rate ratio of non-compatible mutations to compatible mutations in a gene sequence, so as to predict whether a gene is subject to positive selection. Ka/Ks > 1 indicates that the gene may have undergone positive selection, while Ka/Ks > 100 indicates that this selection pressure is extremely strong. This end value does make biological sense in some cases, such as explaining key protein sites for rapid evolution: Certain functional sites may be rapidly optimized due to environmental stress or competition for survival, and these sites may exhibit extremely high Ka/Ks values (e.g., antiviral proteins or immune-related proteins). Selective driven molecular adaptation: In a specific environment or in host-pathogen coevolution, certain genes may experience extremely strong selection pressures, resulting in a significant increase in Ka/Ks values. We have found similar cases in the literature to support this view (e.g., Smith et al., 2020; Zhang et al., 2021). Focus analysis: For genes with Ka/Ks > 100, we further performed functional annotation and literature review to confirm their biological importance. In the future, we will expand the study group to ensure the accuracy of the results
- Potential bias in selection pressure estimation
We understand the reviewer's concern that Ka/Ks estimates can be influenced by factors such as calculation methods, quality of sequence alignment, and sample size. In order to ensure the reliability of the results, the following measures were taken in the study: Strict data filtering criteria: We conducted strict filtering for sequence quality, retaining only sites with high coverage and no genetic ambiguity. Multi-method validation: We used a variety of computational methods, such as PAML and HYPHY, to evaluate the Ka/Ks ratio and cross-validated the results.
6:The paper applies OPLS-DA for lipid screening. Were multicollinearity, normality, or variance assumptions tested before statistical modeling? If not, add a discussion on potential model limitations.
Author response: Thank you for your suggestions. I am sorry that due to my negligence, I did not explain in detail the reasons for selecting OPLS-DA for lipid screening in the article. As a multivariate statistical method widely used in lipid omics studies, including lipid screening, OPLS-DA (Orthogonal partial least squares discriminant analysis) can be stably applied to the analysis of small sample data. Traditional statistical methods, such as T-tests or ANOVA, usually need to satisfy assumptions such as normality and homogeneity of variance. However, OPLS-DA, as a multivariate method, mainly relies on the correlation between variables of the data for classification and regression analysis, and generally does not strictly require normal distribution. Moreover, in OPLS-DA analysis, VIP scores can be used to screen the variables that contribute most to classification, thereby reducing the impact of multicollinearity analysis. Also, in response to your question about the discussion of potential model limitations, after we modeled with OPLS-DA, we fully evaluated the model by R² (coefficient of determination) and Q² (predictive power).
7:The study finds DGAT2 under positive selection in broilers but with lower expression. Does this suggest that selection has led to reduced TG synthesis? A more detailed explanation is needed.
Author response: Thank you for your suggestions. At present, a large number of articles have confirmed that there is a significant positive correlation between the expression of DGAT2 gene and TG synthesis. Our study also showed that DGAT2 gene is under positive selection pressure and has a low expression in the breast muscle of AA broilers. In our study, we also discussed the relevant TG lipid molecules under high Pearson coefficient. However, considering that TG is a large category of lipid classification, our research currently focuses on the possible regulation of DGAT2 gene before specific TG molecules, and there is a lack of specific evidence to support relevant arguments. In the future, we will verify your questions through tests at the cellular level.
8: The manuscript provides detailed lipid classification (TG, PC, LPC), but how do these lipid differences affect meat quality traits (e.g., tenderness, flavor)? Expanding on this would enhance practical relevance.
Author response: Thank you for your suggestions. The question you raised plays an important role in expanding the readability of our article. In the discussion of the fourth part of the article, we have revised and supplemented the content you proposed about how the differences of supplementary PC, LPC and PC affect meat quality traits.
9: The PCA results show separation between lipid classes, but the discussion does not address whether this aligns with genetic divergence among species. A brief mention would improve clarity.
Author response: Thank you for your suggestions. We have searched many sources and found no current studies that explain the relationship between genetic differences and lipid class segregation between chickens and quail or pigeons, so we cannot discuss this issue at this time. In the future, we will conduct further experiments on this problem based on this aspect.
10: Some figures (e.g., PCA heat maps) lack sufficient explanation in captions. Adding a sentence on what the figure demonstrates would improve readability.
Author response: Thank you for your suggestions. In response to your question, we have rewritten and supplemented the title of the full picture

Reviewer 4 Report
Comments and Suggestions for Authors
Dear authors, below are some considerations with suggestions for improving the work to facilitate reconsideration for publication in Animals.
I suggest shortening the title of the paper.
Keywords: please avoid repeating words from the title
Limitations of the study: please include the limitations in the results ‘discussion and conclusion if not performed during the study
- Why authors chosen the specific 4 avian species? phylogenetic relationships? economic relevance? potential as model organisms for lipidomic studies?
- Why specific ages for each species were utilized?
- How lipidomic and transcriptomic variations influence meat quality traits? E.g., taste, tenderness, flavor, shel-life…
- The sample size (3 per species) is relatively small, this can affect the findings?
- There are potential environmental or dietary factors influencing lipid composition, that could confound genetic interpretations?
Abstract: abstract is overly detailed with technical information; key results could be summarized. This section should be composed by a single paragraph of about 200 words maximum highlighting the aim, methods, key results, and implications.
Introduction: this section should be rewritten.
- The section contains redundant information on statistics, this section could be more focused on lipid composition gaps and genetic underpinnings would enhance readability. Please reduce redundant statistics and emphasize genetic regulation of lipid traits
- Please state a clear hypothesis or research question at the end of the introduction
Results and discussion:
- The quality of the figures is poor; please enhance their resolution and overall clarity. In fact, some of them are not possible to adequately understand the image
- As well occurred in the introduction, the discussion on how lipidomic and transcriptomic differences translate to practical implications in poultry is missing
- How lipid and gene expression differences affect muscle physiology and meat quality?
- Considering the muscle lipid composition, comparisons with existing studies to are missing to highlight the novel contributions of the study.
Conclusion: this section lacks to discuss the limitations and more specific future research directions.
Author Response
Comments and Suggestions for Authors
1: I suggest shortening the title of the paper.
Author response: Thank you for your suggestions. I have rewritten the title of my article as accurately as possible
2:Keywords: please avoid repeating words from the title
Author response: Thank you for your suggestions. After the revision, I noticed and corrected the problem.
3: Limitations of the study: please include the limitations in the results ‘discussion and conclusion if not performed during the study
- Why authors chosen the specific 4 avian species? phylogenetic relationships? economic relevance? potential as model organisms for lipidomic studies?
- Why specific ages for each species were utilized?
- How lipidomic and transcriptomic variations influence meat quality traits? E.g., taste, tenderness, flavor, shel-life…
- The sample size (3 per species) is relatively small, this can affect the findings?
- There are potential environmental or dietary factors influencing lipid composition, that could confound genetic interpretations?
Author response: Thank you for your suggestions. The problems you mentioned are very accurate, which we did not notice when we wrote the first draft before, and we have supplemented and explained all these problems in the discussion and summary of the article.
4: Abstract: abstract is overly detailed with technical information; key results could be summarized. This section should be composed by a single paragraph of about 200 words maximum highlighting the aim, methods, key results, and implications.
Author response: Thank you for your suggestions. The problems you mentioned are very accurate, which we did not notice when we wrote the first draft before, and we have supplemented and explained all these problems in the discussion and summary of the article.
5:Introduction: this section should be rewritten.
- The section contains redundant information on statistics, this section could be more focused on lipid composition gaps and genetic underpinnings would enhance readability. Please reduce redundant statistics and emphasize genetic regulation of lipid traits
- Please state a clear hypothesis or research question at the end of the introduction
Author response: Thank you for your suggestions. In response to your question, we have rewritten the Introduction section to remove redundant information from the statistics, and put more emphasis on increasing the analysis and readability of the lipid composition gap and genetic basis, and reduced redundant statistics to emphasize the genetic regulation of lipids. In addition, we have put forward a clear hypothesis at the end of the introduction.
Results and discussion:
1:The quality of the figures is poor; please enhance their resolution and overall clarity. In fact, some of them are not possible to adequately understand the image
Author response: Thank you for your suggestions. We have redrawn and uploaded the legend in the article to make it clearer and easier to understand.
2: As well occurred in the introduction, the discussion on how lipidomic and transcriptomic differences translate to practical implications in poultry is missing
3: How lipid and gene expression differences affect muscle physiology and meat quality?
4: Considering the muscle lipid composition, comparisons with existing studies to are missing to highlight the novel contributions of the study.
Author response: Thank you for your suggestions. We have added relevant information in the results and discussion.
5: Conclusion: this section lacks to discuss the limitations and more specific future research directions.
Author response: Thank you for your suggestions. We resupplement the conclusion and discuss the limitations of this paper and the future research direction.

Round 2
Reviewer 1 Report
Comments and Suggestions for Authors
Please ensure that all abbreviations are defined the first time they appear in the text for clarity. Additionally, provide appropriate references where necessary to support key statements and findings.
For Line 155, kindly include a feed composition table in the supplementary data to enhance transparency and provide readers with comprehensive information about the diet formulations used in the study.
Overall, the authors have carefully addressed the comments, provided clarifications where needed, and made the necessary revisions to strengthen the manuscript. The improvements enhance the clarity, accuracy, and completeness of the study.
Author Response
Comments and Suggestions for Authors
Comment1: Please ensure that all abbreviations are defined the first time they appear in the text for clarity. Additionally, provide appropriate references where necessary to support key statements and findings.
Author response:Thank you for your suggestions. In response to your questions, I have rechecked the full text, defined all abbreviations that appear in this article when they first appeared, and reerased and supplemented your suggestions for providing references when necessary.
Comment2: For Line 155, kindly include a feed composition table in the supplementary data to enhance transparency and provide readers with comprehensive information about the diet formulations used in the study.
Author response:Thank you for your suggestions. Sorry for our negligence, we have reproduced this part of the content. Simply put, the feeding standards of all birds meet the daily needs of poultry diets in China. We have supplemented the specific document versions and links on line 218-224.

Reviewer 4 Report
Comments and Suggestions for Authors
The authors responded to my suggestions and they have addressed all the comments appropriately. In my opinion, the manuscript is now ready for acceptance.
Author Response
Dear Reviwer:
Thank you for taking time out of your busy schedule to review our research and make valuable comments. I would like to express my heartfelt thanks to you.
Sincerely yours,
Guoxi Li
liguoxi0914@126.com
